



# Effects of 2010-2045 climate change on ozone levels in China under carbon neutrality scenario: Key meteorological parameters and processes

Ling Kang[1], Hong Liao[1*], Ke Li[1], Xu Yue[1], Yang Yang[1], Ye Wang[2]

[1] Jiangsu Key Laboratory of Atmospheric Environment Monitoring and Pollution Control, Jiangsu Collaborative Innovation Center of Atmospheric Environment and Equipment Technology, School of Environmental Science and Engineering, Nanjing University of Information Science & Technology, Nanjing 210044, China
[2] Key Laboratory of Meteorological Disaster, Ministry of Education (KLME)/Joint International Research Laboratory of Climate and Environment Change (ILCEC)/ Collaborative Innovation Center on Forecast and Evaluation of Meteorological Disasters (CIC-FEMD), Nanjing University of Information Science and Technology, Nanjing, China,

*Correspondence to*: Hong Liao (hongliao@nuist.edu.cn)

**Abstract.** We examined the effects of 2010-2045 climate change on ozone ($O_3$) levels in China under carbon neutrality scenario using the Global Change and Air Pollution version 2.0 (GCAP 2.0). In eastern China (EC), GCAP 2.0 and other six models from Coupled Model Intercomparison Projection Phase 6 (CMIP6) all projected increases in daily maximum 2-m temperature (T2max), surface incoming shortwave radiation (SW), and planet boundary layer height, and decreases in relative humidity (RH) and sea level pressure. Future climate change is simulated by GCAP 2.0 to have large effects on $O_3$ even under carbon neutrality pathway, with summertime regional and seasonal mean MDA8 $O_3$ concentrations increased by 2.3 ppbv (3.9%) over EC, 4.7 ppbv (7.3%) over North China Plain, and 3.0 ppbv (5.1%) over Yangtze River Delta. Changes in key meteorological parameters were found to explain 58-76% of the climate-driven MDA8 $O_3$ changes over EC. The most important meteorological parameters in summer are T2max and SW in northern and central EC and RH in southern EC. Analysis showed net chemical production was the most important process that increases $O_3$, accounting for 34.0-62.5% of the sum of all processes within the boundary layer. We also quantified the uncertainties in climate-induced MDA8 $O_3$ changes by using CMIP6 multi-model projections of climate and a stepwise multiple linear regression model. GCAP 2.0 results are in the lower-end of the climate-induced increases in MDA8 $O_3$ from the multi-models. These results have important implications for policy-making regarding emission controls under the background of climate warming.

## 1 Introduction

Tropospheric ozone ($O_3$) is a major secondary gas pollutant produced by the complicated photochemical reactions of methane ($CH_4$), carbon monoxide (CO), volatile organic compounds (VOCs) and nitrogen oxides ($NO_x$) in the presence of sunlight. It has adverse effects on human health (Lu et al., 2020; Li et al., 2021; Hong et al., 2019; Dang and Liao, 2019a), ecosystem (Yue et al., 2017; Grulke and Heath, 2020; Ainsworth et al., 2020), and climate (Checa-Garcia et al., 2018; Dang



and Liao, 2019a). Chinese government has implemented the Air Pollution Prevention and Control Action Plan since 2013,
leading to large decline in $NO_x$ emissions and $PM_{2.5}$ concentrations (Zheng et al., 2018; Zhai et al., 2019), but $O_3$ pollution in
eastern China (EC) became worse over the same time period (Tang et al., 2022; Li et al., 2020; Gong et al., 2020; Dang et al.,
2021). Ozone pollution was particularly severe in the North China Plain (NCP), and observed summer mean maximum daily
8-h average (MDA8) $O_3$ concentrations increased at a rate of 3.3 ppb $yr^1$ in NCP from 2013 to 2019, and reached 83 ppb by
2019 (Li et al., 2020). Therefore, it is worth paying attention to the mid-to-long-term changes in $O_3$ concentrations in China in
the future.
The projections of future climate or air quality rely on the future emission pathways under different socioeconomic
scenario assumptions. Shared Socioeconomic Pathways (SSPs) are the state-of-the-art global emission scenarios, which
combines socioeconomic and technological development with future climate radiative forcing outcomes into a scenario matrix
architecture (Gidden et al., 2019). Gidden et al. (2019) constructed nine scenarios of future emissions trajectories, including
SSP1-1.9, SSP1-2.6, SSP2-4.5, SSP3-7.0, SSP3-LowNTCF, SSP4-3.4, SSP4-6.0, SSP5-3.4-Overshoot (OS), and SSP5-8.5.
Among all scenarios, only the SSP1-1.9 scenario achieves net negative emissions of carbon dioxide ($CO_2$) for China and the
world by 2060 (Gidden et al., 2019; Wang et al., 2023), and thus we defined it as the carbon neutrality scenario and applied in
this work. The SSPs scenarios are used in Scenario Model Intercomparison Project (ScenarioMIP) in Coupled Model
Intercomparison Projection Phase 6 (CMIP6) to facilitate the integrated analysis of future climate impacts, vulnerabilities,
adaptation, and mitigation (Gidden et al., 2019; Riahi et al., 2017).
Future $O_3$ concentrations depend on the future emissions. Shi et al. (2021) projected the $O_3$ concentration changes in
China over 2020-2060 with no changes in meteorological conditions based on the Chinese Academy of Environmental
Planning Carbon and Air Quality Pathways (CAEP-CAP) for pursuing the carbon neutrality. The 90[th] percentile of daily
maximum 8-h average (MDA8) $O_3$ (90[th] MDA8 $O_3$) in China reduced from 138 μg $m^{-3}$ in 2020 to 93 μg $m^{-3}$ in 2060 (a 84%
reductions in 90[th] MDA8 $O_3$). Based on Ambitious-pollution-Neutral-goals scenario from the Dynamic Projection model for
Emissions in China (DPEC), Xu et al. (2022) used a regional climate-chemistry-ecology model to assess the impacts of regional
emission reductions in China with the goal of achieving carbon neutrality by 2060, and found that the national average annual
$O_3$ concentrations would decline by 35.6 μg $m^{-3}$ over 2015-2060. Wang et al. (2023) reported by using the GEOS-Chem model
that the $O_3$ levels in Beijing-Tianjin-Hebei Region (BTH), Yangtze River Delta Region (YRD), Pearl River Delta Region
(PRD), Sichuan Basin Region (SCB), and Fenwei Plain (FWP) under SSP1-1.9 scenario could meet the air quality standard
by 2030, while those under SSP5-8.5 could not meet even by 2060. The 90[th] MDA8 $O_3$ in BTH, YRD, PRD, SCB, and FWP
during 2015-2060 would change by -27.3%, -27.6%, -33.1%, -33.1%, and -31.8% under SSP1-1.9 scenario, and by +8.6%,
+7.6%, +5.2%, -0.5%, and +2.9% under SSP5-8.5 scenario (Wang et al., 2023), respectively. However, these studies did not
examine the effects of future climate change on $O_3$ concentrations.
Future $O_3$ concentrations also depend on future climate. Using the Weather Research and Forecasting Model with
Chemistry (WRF-Chem) driven by Community Climate System Model version 3 (CCSM3), Liu et al. (2013) predicted that
climate change caused a 1.6 ppb increase in surface $O_3$ over South China in October 2000-2050 under the IPCC A1B scenario.



They show that future elevated near-surface temperature (1.6 °C) and increased emissions of isoprene (5-55%) and
monoterpenes (5-40%) would lead to increases in chemical production of $O_3$. By using GEOS-Chem model driven by NASA
Goddard Institute for Space Studies (GISS) general circulation model (GCM) 3 under the A1B scenario, Wang et al. (2013)
reported that climate change would cause a 0.55 ppbv increase in annual mean surface $O_3$ in EC over 2000-2050, in which
more than 40% could be attributed to climate-induced increases in biogenic VOCs (BVOCs) emissions. Climate-induced
increases in $O_3$ levels over EC are most pronounced and spatially extensive in summer, with a summer-average of 1.7 ppbv
and a maximum of 10 ppbv. By employing a combination of models, Hong et al. (2019) projected that warm-season (April-
September) averages of daily 1-h maximum $O_3$ levels would increase by 2-8 ppb in most of EC from 2006–2010 to 2046–
2050 under the Representative Concentration Pathway 4.5 (RCP4.5), in which 14% could be attributable to increased future
heat wave days. Based on sensitivity simulations from five CMIP6 models by fixing sea surface temperatures (SSTs) at present-
day or future conditions in the SSP3-7.0 scenario, Zanis et al. (2022) reported that the sensitivity of $O_3$ to temperature would
enhance in regions close to anthropogenic sources or BVOCs emission sources (e.g., southern EC), with the values ranging
from 0.2 to 2 ppbv °$C^{-1}$. However, the scenarios utilized in these studies are not the representative scenarios in China in the
context of carbon neutrality.
Few studies have examined the impacts of climate change under low-carbon or carbon-neutrality scenario. Li et al. (2023)
showed that the annual mean surface $O_3$ during 2025-2095 increased by 0-2 ppb over EC under the SSP1-2.6 scenario by using
a machine learning (ML) model along with multi-source data,with reduced relative humidity and enhanced downward solar
radiation in the future favouring photochemical formation of surface $O_3$. Zhu et al. (2024) investigated the effects of global
and regional SSTs changes on surface $O_3$ levels in China during the warm season in 2050 (averaged over 2045-2054) based
on global chemistry model simulations. They found that, compared with SSP5-8.5 scenario, future cooling of global ocean,
North Pacific Oceans, and Southern Hemisphere oceans in SSP1-1.9 scenario would contribute to 0.79, 0.48, and 0.58 ppbv
decreases in surface $O_3$ concentrations over EC, respectively, as a result of the weakened chemical production and anomalous
upward airflow. However, these studies did not quantify the impacts of the dominant meteorological parameters and processes.
Climate change can influence tropospheric $O_3$ through altering meteorological fields and meteorology-sensitive physical
and chemical processes. Integrated process rate (IPR) analysis, multiple linear regression (MLR) model and Lindeman,
Merenda, and Gold (LMG) method are widely used to examine the contributions of main processes and key meteorological
parameters to $O_3$ changes in China (Gong et al., 2022; Dang et al., 2021; Li et al., 2019). Liu et al. (2013) found that climate-
induced changes in boundary layer $O_3$ budget were dominated by chemical processes, with gas-phase chemical reaction yield
increasing by 3ppb $h^{-1}$ in PRD over 2000-2050. The maximum increases in $O_3$ by chemical process were located in areas with
significant warming as well as high anthropogenic and biogenic emissions of precursors. By combining MLR model and LMG
method, Dang et al. (2021) showed that higher temperature and anomalous southerlies were key meteorological contributors
to summer $O_3$ increases in NCP in 2017 relative to 2012, while weaker wind speeds and lower relative humidity were the key
contributors in YRD. Gong et al. (2022) found by using the IPR analysis that net chemical production, diffusion, dry deposition,





horizontal advection and vertical advection during $O_3$ pollution events in 2014-2017 changed by 3.3, -1.1, -0.4, -9.1 and 8.1
Gg $O_3$ $d^{-1}$ in North China relative to the seasonal mean values. The positive effects of net chemical production and vertical
advection were associated with a typical weather pattern characterized by high daily maximum temperatures, low relative
humidity, anomalous southerlies and divergence in the low troposphere, and anomalous downward airflow from 500 hPa to
the surface. However, to our knowledge, no study has combined these approaches to quantify the roles of key meteorological
parameters and associated processes in climate-induced changes in tropospheric $O_3$ levels in China under the carbon neutrality
scenario.
In this study, based on the version 2.0 of the Global Change and Air Pollution (GCAP 2.0) model framework, we examine
the effects of 2010-2045 climate change on $O_3$ levels in China under carbon neutrality scenario, focusing on the key
meteorological parameters and processes for climate-induced $O_3$ changes by using the stepwise MLR model, LMG method
and IPR analysis. The observations and CMIP6 data, numerical models and experiments, and statistical analysis methods are
given in Sect. 2. Section 3.1 shows GCAP 2.0 projected climate change over 2010-2045 and the comparison with other six
CMIP6 model projections. Simulated present-day $O_3$ concentrations and model evaluation, and future tropospheric $O_3$ changes
driven by 2010-2045 climate change are presented in Sect. 3.2. Section 3.3 quantifies the key meteorological parameters and
processes for climate-induced $O_3$ changes. The climate-driven MDA8 $O_3$ changes predicted by stepwise MLR model using
climate outputs from CMIP6 models are shown in Sect. 3.4. Section 3.5 examines briefly the effects of emission change alone
on $O_3$ levels. The conclusions are presented in Sect. 4.
**2 Data and methods**
**2.1 Observations**
The real-time monitoring air quality data released by the China National Environmental Monitoring Center (CNEMC)
became operational in 2013. $O_3$ concentrations are measured by the ultraviolet spectrophotometry method, following the China
Environmental           Protection           Standards           'HJ           654-2013'
(https://www.mee.gov.cn/ywgz/fgbz/bz/bzwb/jcffbz/201308/W020130802491142354730.pdf).   We   used   hourly   $O_3$
concentrations at 1479 sites nationwide in 2015 and converted the data unit from micrograms per cubic meter (μg $m^{-3}$) to parts
per billion per volume (ppbv). Data quality control went through the following steps: (1) negative or missing values were
removed; (2) MDA8 $O_3$ concentration was calculated if there were at least 6 hours of valid data in each 8-hour period; (3) a
site with more than 95% valid data in 2015 was retained (1047 sites after data quality control). For model evaluation, observed
MDA8 $O_3$ concentrations were averaged over sites within each of the 2° latitude by 2.5° longitude model grid cell (with a total
of 118 grids).



## 2.2 Numerical models and experiments

### 2.2.1 GCAP 2.0 model framework

GCAP 2.0 model framework is a one-way offline coupling between the version E2.1 of the NASA Goddard Institute for Space Studies (GISS-E2.1) GCM and the global 3-D chemical transport model GEOS-Chem (Murray et al., 2021). Both the GISS-E2.1 GCM and the GEOS-Chem models have a horizontal resolution of 2° latitude by 2.5° longitude with 40 vertical layers extending from the surface to 0.1 hPa.

GISS-E2.1 GCM participated in CMIP6 experiments was described in detail by Kelley et al. (2020) and Miller et al. (2021). GISS-E2.1 contributed several configurations to CMIP6, and Murray et al. (2021) used the atmosphere-only configuration with the prescribed sea surface temperatures to re-perform the simulation of "r1i1p1f2" variant label and archived the subdaily meteorological diagnostics necessary for driving GEOS-Chem, namely GCAP 2.0 meteorology. The GCAP 2.0 meteorology (http://atmos.earth.rochester.edu/input/gc/ExtData/GCAP2/CMIP6/) for driving GEOS-Chem model (version 13.2.1, http://wiki.seas.harvard.edu/geos-chem/index.php/GEOS-Chem_13.2.1) only covered the periods of the pre-industrial era (1851-1860), the recent past (2001-2014), the near-future (2040-2049), and the end-of-the-century (2090-2099) for seven future scenarios.

Version 13.2.1 of the GEOS-Chem model has $O_x$-$NO_x$-hydrocarbon-aerosol tropospheric chemistry mechanism (Bey et al., 2001; Pye et al., 2009) with the updated stratospheric chemistry mechanism from NASA's Global Modeling Initiative (GMI). Photolysis rates are calculated based on Fast-JX v7.0 scheme (Eastham et al., 2014; Jiang et al., 2013). Aerosols influence tropospheric $O_3$ through heterogeneous reactions and the changes in photolysis rates (Lou et al., 2014; Li et al., 2019). Dry deposition is computed using a resistance-in-series model (Wesely, 1989) with a number of modifications (Wang et al., 1998). Vertical mixing in planetary boundary layer (PBL) is calculated by a nonlocal scheme (Lin and Mcelroy, 2010). Cloud convection is parameterized as a single plume acting under the mean upward convective, entrainment, and detrainment mass for each level of a model column as archived from the GCM (Murray et al., 2021).

### 2.2.2 Emissions

The available emission years of SSPs inventory are 2015, 2020, 2030, 2040, 2050, 2060, 2070, 2080, 2090, and 2100. Therefore, corresponding to the mid-term climate change, we chose 2015 and 2050 emissions to represent the present-day and future emissions, respectively. Present-day (year 2015) and future (year 2050) anthropogenic and biomass burning emissions are given in Table 1. Year 2050 anthropogenic and biomass burning emissions are based on the SSP1-1.9 scenario of CMIP6 experiments. The anthropogenic and biomass burning emissions of $NO_x$, CO, and NMVOCs are 27.2, 161.8, and 24.8 Tg yr$^{-1}$ in EC in 2015, respectively, and are projected to decrease by 80.0%, 63.2%, and 70.0% in 2050 relative to 2015, respectively. These changes are larger than the decreases in global total emissions (64.1%, 52.3%, and 31.6%, respectively). The anthropogenic emissions of sulfur dioxide ($SO_2$), organic carbon (OC), and black carbon (BC) are projected to decrease by





95.3%, 67.1%, and 84.8% in EC, and by 79.9%, 69.1%, and 82.6% globally, respectively, while ammonia ($NH_3$) emission
remains stable.
Table 1 also lists the climate-sensitive natural emissions, including lightning and soil emissions of $NO_x$ and biogenic
emissions of VOCs which are calculated online based on the GCAP 2.0 meteorology. Lightning and soil emissions of NOx
are calculated using the cloud-top height scheme of Price and Rind (1992) and the Berkeley-Dalhousie Soil $NO_x$
Parameterization (BDSNP) scheme developed by Hudman et al. (2012), respectively. Biogenic VOCs (BVOCs) emissions are
computed using the Model of Emissions of Gases and Aerosols from Nature Version 2.1 (MEGAN v2.1) (Guenther et al.,
2012). In present-day, the lightning and soil emissions of $NO_x$ and biogenic emissions of VOCs are 0.6, 1.1, and 16.0 Tg $yr^{-1}$
in EC, respectively. Note that VOCs from the biogenic sources (16.0 Tg $yr^{-1}$) are comparable to those from the anthropogenic
emissions (24.4 Tg $yr^{-1}$) in EC. Compared to 2015, lightning and soil emissions of $NO_x$ and the BVOCs emissions are predicted
to increase by 8.8%, 5.6 %, and 15.5% in EC, respectively.
**Table 1. The annual anthropogenic, biomass burning, and natural emissions (Tg $yr^{-1}$) for the present-day (year 2015) and the future**
**(year 2050) under SSP1-1.9 scenario. The domain of eastern China (EC) is 21.00° – 45.00° N, 106.25° – 123.75° E.**

|  |  | Global | | | Eastern China | | |
| --- | --- | --- | --- | --- | --- | --- | --- |
|  |  | 2015 | 2050 | Change (%) | 2015 | 2050 | Change (%) |
| $NO_x$ | Anthropogenic | 119.82 | 36.27 | -69.73 | 27.14 | 5.38 | -80.18 |
|  | Biomass burning | 13.74 | 11.72 | -14.70 | 0.07 | 0.06 | -14.29 |
|  | Lightning | 20.25 | 21.13 | 4.35 | 0.57 | 0.62 | 8.77 |
|  | Soil | 35.64 | 36.98 | 3.76 | 1.08 | 1.14 | 5.56 |
| CO | Anthropogenic | 608.00 | 188.74 | -68.96 | 159.61 | 57.69 | -63.86 |
|  | Biomass burning | 328.44 | 258.18 | -21.39 | 2.19 | 1.81 | -17.35 |
| NMVOCs | Anthropogenic | 284.21 | 189.46 | -33.34 | 24.41 | 7.14 | -70.75 |
|  | Biomass burning | 49.11 | 38.35 | -21.91 | 0.34 | 0.28 | -17.65 |
|  | Biogenic VOCs | 941.17 | 1029.46 | 9.38 | 15.95 | 18.42 | 15.49 |
| $SO_2$ | Anthropogenic | 98.63 | 19.87 | -79.85 | 20.67 | 0.98 | -95.26 |
|  | Biomass burning | 2.16 | 1.75 | -18.98 | 0.02 | 0.01 | -50.00 |
| $NH_3$ | Anthropogenic | 61.34 | 61.73 | 0.64 | 7.65 | 7.71 | 0.78 |
|  | Biomass burning | 3.91 | 2.97 | -24.04 | 0.03 | 0.03 | 0.00 |
| OC | Anthropogenic | 19.59 | 6.05 | -69.12 | 4.26 | 1.40 | -67.14 |
|  | Biomass burning | 15.23 | 11.34 | -25.54 | 0.12 | 0.09 | -25.00 |
| BC | Anthropogenic | 7.99 | 1.39 | -82.60 | 2.10 | 0.32 | -84.76 |
|  | Biomass burning | 1.75 | 1.41 | -19.43 | 0.01 | 0.01 | 0.00 |





**2.2.3 Numerical experiments**
Considering the available GCAP 2.0 meteorology, 2005-2014 meteorology is used to represent the present-day climate
(2010), and 2040-2049 meteorology under SSP1-1.9 scenario is used to represent the future climate (2045). To examine the
respective and combined effects of future changes in climate and emissions on surface $O_3$ levels, four numerical experiments
are set up (Table 2). The simulations of CpdEpd, CpdEfut, CfutEpd, and CfutEfut represent, respectively, $O_3$ levels under
present-day climate and emissions, present-day climate and future emissions, future climate and present-day emissions, and
future climate and emissions. Therefore, (CfutEpd minus CpdEpd) or (CpdEfut minus CpdEpd) indicates the individual effect
of climate change or emission change on $O_3$ concentrations, and (CfutEfut minus CpdEpd) indicates the combined effect of
climate and emission changes. To smooth out the noise of natural climate variabilities, each simulation is conducted for 10
years after a 1-year spin-up. Unless otherwise noted, all the results presented in this study are 10 yr averages of 2005-2014 or

182 2040-2049.

**Table 2. Experiment design.**

| Description | Meteorological fields | Natural emissions | Anthropogenic emissions | Biomass burning emissions |
|---|---|---|---|---|
| CpdEpd | 2005-2014 | 2005-2014 | 2015 | 2015 |
| CpdEfut | 2005-2014 | 2005-2014 | 2050 | 2050 |
| CfutEpd | 2040-2049 | 2040-2049 | 2015 | 2015 |
| CfutEfut | 2040-2049 | 2040-2049 | 2050 | 2050 |

**2.3 Statistical analysis methods**
**2.3.1 Stepwise MLR model and LMG method**
To identify meteorological variables that have a significant effect on climate-induced MDA8 $O_3$ changes, we applied
stepwise multiple linear regression (MLR) model to relate 10 yr daily MDA8 $O_3$ anomalies to 10 yr daily meteorological
parameter anomalies in the target region or each grid cell. The time series of 10 yr daily MDA8 $O_3$ anomalies are obtained by
(CfutEpd minus CpdEpd), and 10 yr daily meteorological parameter anomalies are obtained by subtracting 2005-2014 from
2040-2049. Nine meteorological variables are considered in the MLR analysis (Table 3), including daily maximum 2-m air
temperature (T2max), relative humidity (RH), surface incoming shortwave radiation (SW), planet boundary layer height
(PBLH), precipitation (PREC), sea level pressure (SLP), and 850 hPa wind fields (U850, V850, and WS850). We first
correlated 10 yr daily MDA8 $O_3$ anomalies with 10 yr daily meteorological parameter anomalies, and excluded meteorological
variables that are not significantly correlated with MDA8 $O_3$ at the 95% confidence level. We then performed collinearity
statistics on the retained meteorological variables based on the variance inflation factor (VIF): the meteorological variable with





the largest VIF was sequentially excluded until the VIFs of all meteorological variables were less than 10. After these steps,
the reserved meteorological variables were read into the stepwise MLR model, which is in the following form (Li et al., 2019):
$y = \beta_0 + \sum_{k=1}^{N} \beta_k x_k + interaction\ term$ ,              (1)
where $y$ is the daily MDA8 $O_3$ anomalies, $(x_1, \dots, x_N)$ are the $N$ meteorological variable screened by stepwise MLR model,
and $\beta_k$ is the regression coefficient for the $k$-th meteorological variable. The adjusted coefficient of determination ($R^2$_adj) of
MLR equation represents the proportion of climate-induced MDA8 $O_3$ changes that can be explained by the changes in key
meteorological variables.
We then used the Lindeman, Merenda, and Gold (LMG) method (Grömping, 2006) to quantify the relative contribution
of each meteorological variable reserved in MLR equation. The LMG method decomposes the MLR model-explained total
$R^2$_adj into non-negative individual $R^2$_adj contribution from each correlative regressor.
**Table 3. Meteorological variables considered in the statistical analysis.**

| Abbreviation | Description |
| --- | --- |
| T2max | Daily maximum 2-m temperature (K) [a] |
| RH | Relative humidity (%) [b] |
| SW | Surface incoming shortwave radiation (W m$^{-2}$) [a] |
| PBLH | Planet boundary layer height (m) [a] |
| PREC | Precipitation (mm d$^{-1}$) [a] |
| SLP | Sea level pressure (hPa) [a] |
| U850 | 850 hPa zonal wind (m s$^{-1}$) [b] |
| V850 | 850 hPa meridional wind (m s$^{-1}$) [b] |
| WS850 | 850 hPa wind speed (m s$^{-1}$) [c] |

[a] Temporal resolution is 1-hour
[b] Temporal resolution is 3-hour
[c] Calculated from the horizontal wind vectors (U850, V850).
**2.3.2 IPR analysis**
Integrated process rate (IPR) analysis is used to quantify the contributions of climate-driven change in physical and
chemical processes to $O_3$ mass changes in different seasons in EC (21.00-45.00°N, 106.25-123.75°E). Five processes that
influence $O_3$ levels are investigated, including net chemical production, PBL mixing, dry deposition, cloud convection, and
horizontal and vertical advection transport, which jointly determine the $O_3$ mass balance. All of the processes are diagnosed at
every timestep and then summed over each day. The contribution of each process was calculated following Eqs. (2) and (3)
(Dang and Liao, 2019b):





$$PC_{DIFF\_i} = PC_{\text{CfutEpd}\_i} - PC_{\text{CpdEpd}\_i} \,, \qquad\qquad (2)$$
$$\%PC_{DIFF\_i} = \frac{PC_{DIFF\_i}}{\sum_i^n abs(PC_{DIFF\_i})} \times 100\% \,, \qquad\qquad (3)$$
where $n$ is the number of processes ($n = 5$), $PC_{\text{CpdEpd}\_i}$ and $PC_{\text{CfutEpd}\_i}$ are the seasonal mean $O_3$ mass by process $i$ from the
CpdEpd and CfutEpd simulations, respectively, and $PC_{DIFF\_i}$ is the climate-driven change in $O_3$ mass by process $i$. $\%PC_{DIFF\_i}$
is the proportion of process $i$ in the total $O_3$ mass change caused by all processes. Note that the sum of absolute values of
$\%PC_{DIFF\_i}$ for all processes equals 100%. The IPR analysis method has been widely used in previous studies to identify the
key processes that contribute to air pollution episodes (Gong and Liao, 2019; Dai et al., 2023; Dang and Liao, 2019b) or drive
the interannual and decadal variations in air pollutants (Yang et al., 2022; Mu and Liao, 2014).
**2.4 CMIP6 data**
The projected climate change by GCAP 2.0 may have uncertainties. To identify the range of uncertainties of the effects
of climate change on MDA8 $O_3$, we downloaded multi-model results of monthly means of the meteorological variables
consistent with those in Table 3 in present-day (2005-2014) and future (2040-2049) under SSP1-1.9 scenario from the CMIP6
data repository (https://esgf-node.llnl.gov/search/cmip6/). Since only six climate models in CMIP6 can provide PBLH, we
selected outputs with the "r1" variant label from these models (Table S1). Note that GISS-E2.1-G and GISS-E2.1-H are
coupled models of the GISS-E2.1 atmospheric model with the GISS and HYCOM ocean models, respectively, while the GCAP
2.0 (or GISS-E2.1) is the atmosphere-only model with the prescribed sea surface temperatures. We extracted the monthly
values for 2005-2014 and 2040-2049 from the raw data and interpolated them into GCAP 2.0 resolution (2° × 2.5°) by bilinear
interpolation.
**3 Results**
**3.1 Projected future climate change over China**
**3.1.1 Projected climate change over 2010-2045 by GCAP 2.0**
Figure 1 shows the projected 2010-2045 changes in seasonal mean T2max, RH, SW, PBLH, PREC, U850 and V850, and
SLP in winter (December-January-February, DJF), spring (March-April-May, MAM), summer (June-July-August, JJA), and
autumn (September-October-November, SON) over China by GCAP 2.0 (or GISS-E2.1 GCM) under SSP1-1.9 scenario. The
projected T2max, SW, and PBLH generally increase over EC while RH generally decreases. Regionally, the maximum
increases in T2max occur in the northeastern China in DJF (2.0-2.5 K). NCP (green rectangle in Fig. 1) has the largest
temperature increases in other seasons, with values of 2.0-2.5 K in MAM, 1.5-2.0 K in JJA, and 1.0-1.5 K in SON. RH has a
decrease of 2-6% over northern China in MAM and JJA, and of 2-4% over southern China in SON. Changes in SW and PBLH
have similar spatial distributions, both of which increase largely over northern China in MAM and JJA. Precipitation generally



increases over southeastern China in DJF and SON, and decreases in northern China in MAM. With respect to atmospheric
circulations, over the Northwestern Pacific Ocean, there is an anomalous high-pressure in DJF and an anomalous low-pressure
in other seasons. As a result, over EC, anomalous southerlies prevail in DJF and anomalous northwesterlies/northerlies prevail
in other seasons.

**Figure 1. Projected 2010-2045 changes in seasonal mean (a) daily maximum 2-m air temperature (T2max, K), (b) surface relative**
**humidity (RH, %), (c) surface incoming shortwave radiation (SW, W m$^{-2}$), (d) planet boundary layer height (PBLH, m), (e)**
**precipitation (PREC, mm d$^{-1}$), and (f) wind fields at 850 hPa (arrows, m s$^{-1}$) and sea level pressure (SLP, shades, hPa) by GCAP 2.0**
**under SSP1-1.9 scenario. The dotted areas and red arrows represent a statistically significant difference at 95% confidence**
**according to Student's two sample t test. The black, green and blue rectangles in (a) indicate the domain of eastern China (EC, 21.00-**
**45.00°N, 106.25-123.75°E), North China Plain (NCP, 35.00-41.00°N, 113.75-118.75°E), and Yangtze River Delta (YRD, 29.00-33.00°N,**
**118.75-123.75°E), respectively.**



### 3.1.2 Comparisons with projected climate change from other CMIP6 models

The projected 2010-2045 changes in meteorological parameters (Table 3) under SSP1-1.9 scenario over EC by GCAP 2.0 are compared with those from six other CMIP6 models in Fig. 2. Increases in T2max, SW, and PBLH throughout the year are robust features among all CMIP6 models. Most models projected reductions in RH and SLP and increases in PREC. However, there are large model differences in winds at 850 hPa with inconsistent sign of changes. On a multi-model mean (MMM) basis, projected annual mean changes over EC in T2max, SW, PBLH, PREC, RH, and SLP are 1.4 K, 11.8 W m$^{-2}$, 30.6 m, 0.3 mm day$^{-1}$, -0.7%, and -0.3 hPa, respectively. Consistent with the MMM, the GCAP 2.0 projections show overall increases in T2max, SW, PBLH, and PREC and decreases in RH and SLP, with the annual mean changes of 1.1 K, 7.3 W m$^{-2}$, 23.7 m, 0.03 mm day$^{-1}$, -1.3%, and -0.3 hPa, respectively. Therefore, relative to the MMM, GCAP 2.0 underestimates the increases in T2max, SW, PBLH, and PREC and overestimates the decreases in RH. The uncertainties in simulated future O$_3$ caused by the uncertainties in future climate change will be quantified in Sect. 6.

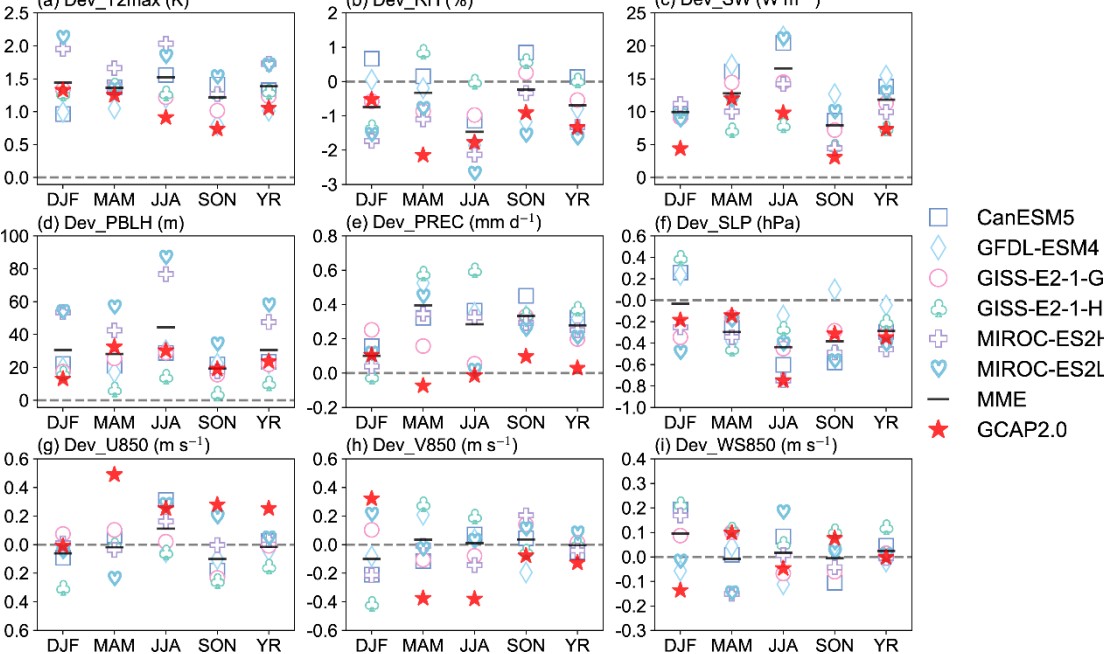

**Figure 2. Comparisons of simulated 2010-2045 changes in seasonal and annual mean meteorological parameters over EC by GCAP 2.0 with those by other six CMIP6 models under SSP1-1.9 scenario. Note that GISS-E2.1-G and GISS-E2.1-H are coupled models of the GISS-E2.1 atmospheric model with the GISS and HYCOM ocean models, respectively, while the GCAP 2.0 (or GISS-E2.1) is the atmosphere-only model with the prescribed sea surface temperatures. The multi-model mean (MMM) is calculated from the average of the six CMIP6 models. Different markers represent different models, black lines represent MMM, and red stars represent GCAP 2.0 results.**



**3.2 Simulated present-day and future tropospheric O₃**

**3.2.1 Present-day tropospheric O₃ and model evaluation**

Figure 3 shows simulated present-day MDA8 O₃ concentrations from CpdEpd simulation and the observations in 2015 from CNEMC. We use 2015 observations to evaluate the simulated present-day MDA8 O₃ concentrations because emissions of year 2015 are used for present-day. Simulated MDA8 O₃ concentrations in EC are highest in JJA (50-70 ppbv), followed by MAM (35-55 ppbv), SON (30-50 ppbv), and DJF (10-45 ppbv). The model generally captures the spatial distributions of the observed seasonal mean MDA8 O₃ levels over China, with spatial correlation coefficients (R) of 0.63, 0.12, 0.54, and 0.33 in DJF, MAM, JJA, and SON, respectively. Dang and Liao (2019a) also reported a low spatial correlation coefficient (R of 0.08) between observed and simulated seasonal mean O₃ in China in MAM of 2014-2017, which was attributed to the negative biases in NCP and YRD whereas the positive biases outside these two regions. The model overestimates MDA8 O₃ concentrations in China, with normalized mean biases (NMBs) of 7.1-18.6% in different seasons. Figure S1 shows monthly variations in simulated and observed MDA8 O₃ levels over EC, NCP, and YRD. Both observed and simulated monthly mean MDA8 O₃ concentrations are high during warm months (April-September) in these three regions. The NMBs in EC, NCP, and YRD are 11.1%, -12.8% and -0.9%, respectively, which is consistent with results of Dang and Liao (2019a). The scattering plots of model results vs. observations for grids in these three regions show correlation coefficients (R) of 0.76 to 0.94 when all of the year 2015 data are considered.



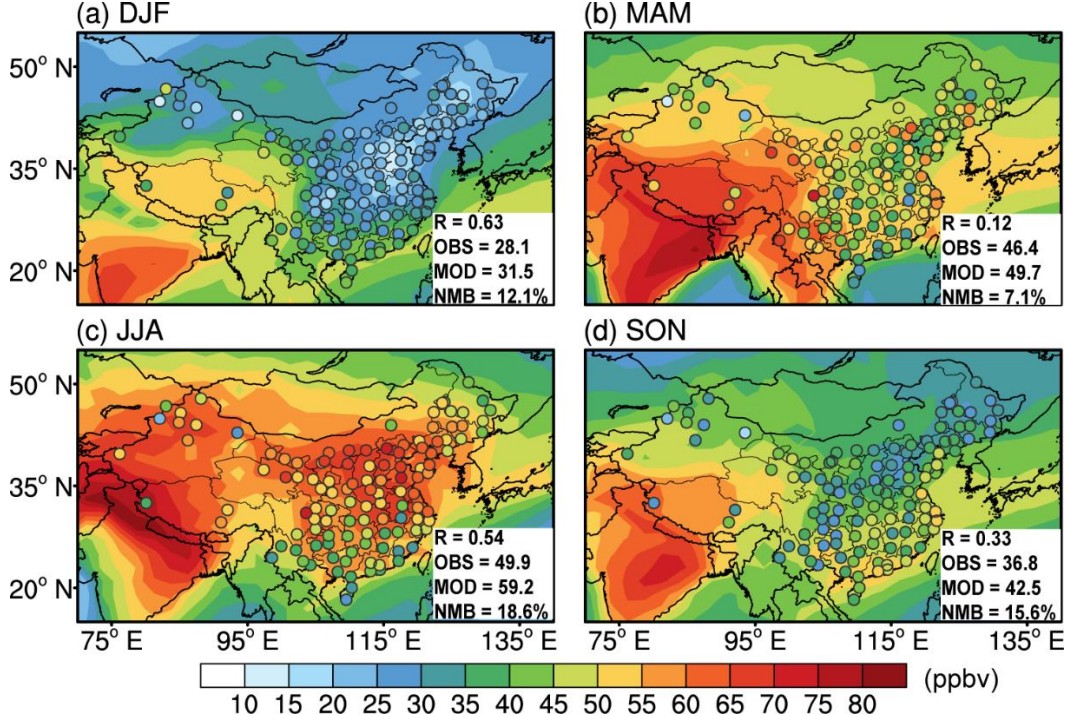

**Figure 3. Spatial distributions of observed (CNEMC, circles) and simulated (CpdEpd, shades) seasonal mean MDA8 O₃ concentrations (ppbv) in 2015. Observed (OBS) and simulated (MOD) values that averaged over 118 grids, and their spatial correlation coefficients (R) and normalized mean biases (NMB) are also shown at the bottom right corner of each panel.**

**3.2.2 Future changes in tropospheric O₃ driven by climate change**

Figure 4a shows future changes in seasonal mean MDA8 O₃ concentrations due to climate change (CfutEpd minus CpdEpd). Climate change alone causes large increases in MDA8 O₃ values over EC in MAM and JJA, and the maximum value reaching 7.6 ppbv in NCP in JJA. In DJF, MAM, JJA, and SON, the regional and seasonal mean MDA8 O₃ values increase by 0.5 (1.5%), 1.3 (2.7%), 2.3 (3.9%), and 0.4 ppbv (1.0%) in EC, by 0.4 (2.0%), 2.8 (6.7%), 4.7 (7.3%), and 1.5 ppbv (4.6%) in NCP, and by 1.1 (3.5%), 1.7 (3.3%), 3.0 (5.1%), and 0.3 ppbv (0.6%) in YRD, respectively.



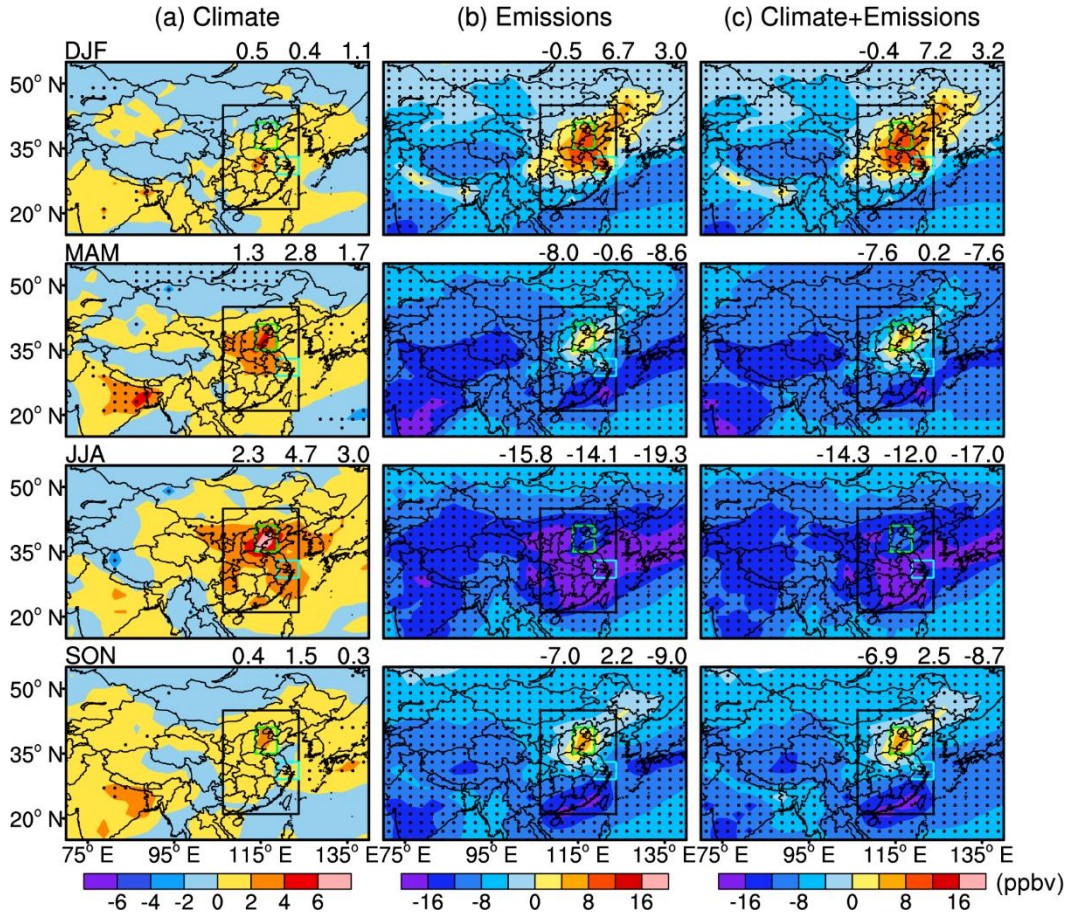

**Figure 4. Predicted future changes in seasonal mean MDA8 O₃ concentrations (ppbv) due to (a) climate change alone (CfutEpd minus CpdEpd), (b) emission change alone (CpdEfut minus CpdEpd), and (c) combined climate and emission changes (CfutEfut minus CpdEpd) under SSP1-1.9 scenario. The black, green and blue rectangles indicate the domain of EC, NCP, and YRD, respectively. The dotted areas represent a statistically significant difference at the 95% level according to Student's two sample *t* test. The values at the top right of each panel are the regional mean values of EC, NCP, and YRD, respectively.**

The pressure-latitude cross sections of climate-driven seasonal mean $O_3$ changes from the surface to 500 hPa for EC, NCP, and YRD are shown in Fig. 5. Vertically, $O_3$ increases of exceeding 1 ppbv extend from the surface to 500 hPa altitude over the three regions in JJA. The maximum $O_3$ increases of 4-5 ppbv in NCP occur both at the surface and around 850 hPa, and those of 3-5 ppbv in the YRD occur between 930 and 736 hPa. The $O_3$ increases over EC is large below 700 hPa over 25-41 °N, and the location of high values shifts from north to south with altitude, which is dominated by the pattern of NCP. In other seasons, the $O_3$ increases of 1-3 ppbv are generally near the surface.





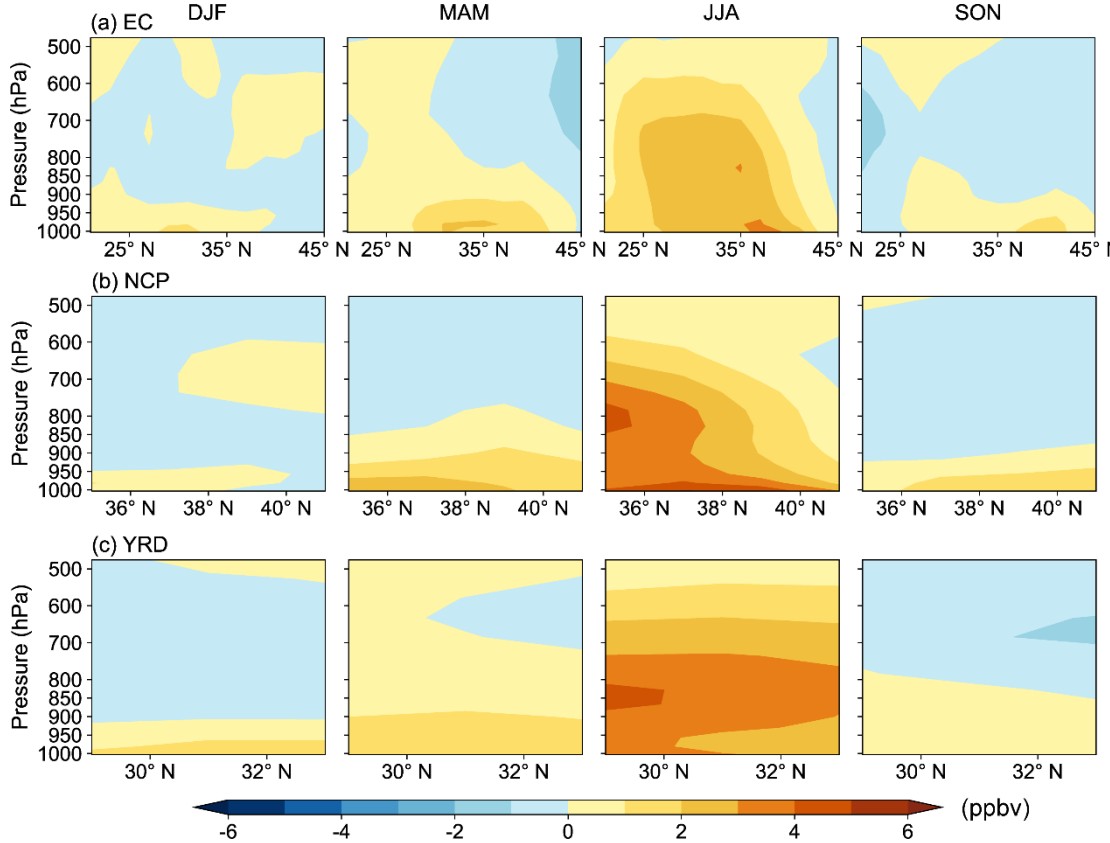


**Figure 5. The pressure-latitude cross sections of climate-driven seasonal mean O₃ changes (ppbv) averaged over the longitudes of (a)**

**106.25-123.75°E for EC, (b) 113.75-118.75°E for NCP, and (c) 118.75-123.75°E for YRD.**
**3.3 Key meteorological parameters and processes for climate-induced O₃ changes**
**3.3.1 Key meteorological parameters for climate-induced MDA8 O₃ changes**
For climate-induced changes in MDA8 O₃, the stepwise MLR model is used to identify key meteorological variables that
have statistically significant effect on MDA8 O₃, and the obtained $R^2$_adj represents the proportion of climate-induced MDA8
O₃ changes that can be explained by the changes in these key meteorological variables retained in MLR equation. Then, the
LMG method decomposes the MLR model-explained total $R^2$_adj and get the relative contribution of each meteorological
variable.
Table 4 shows the MLR equations between the daily anomalies of MDA8 O₃ and daily anomalies of meteorological
variables over EC for each season. The daily anomalies of both MDA8 O₃ and meteorological variables are 10 yr daily values,
which were derived from (CfutEpd minus CpdEpd) and ((2040-2049) minus (2005-2014)), respectively. For each key
meteorological variable, the positive or negative regression coefficient represents statistically significant positive or negative
effect of this variable on MDA8 O₃ concentrations. The $R^2$_adj of the MLR equations are 0.76, 0.74, 0.58, and 0.76 in DJF,




MAM, JJA, and SON, respectively, indicating 76%, 74%, 58%, and 76% of the climate-induced changes in MDA8 $O_3$ can be
explained by the changes in the key meteorological variables retained in MLR equations. Figure 6 shows LMG decomposed
contribution of each key meteorological variable in fitting climate-driven MDA8 $O_3$ changes over EC. The top three important
meteorological variables are T2max, SW, and RH, with the total contributions of 71.2% (T2max + SW + RH) in DJF, 78.2%
(T2max + SW + RH) in MAM, 70.1% (SW + RH + T2max) in JJA, and 49.9% (T2max + RH) in SON. PBLH is also a major
meteorological variable with the contributions of 9.6-24.5% in different seasons. The total contributions of the circulation
changes are 13.4% (SLP + WS850 + V850), 9.8% (V850 + U850), 11.4% (WS850 + V850 + SLP), and 9.5% (SLP + V850 +
WS850) in DJF, MAM, JJA, and SON, respectively.
**Table 4. Stepwise multiple linear regression (MLR) equations between the daily anomalies of MDA8 $O_3$ (CfutEpd minus CpdEpd)**
**and daily anomalies of meteorological parameters in EC. All the regression coefficients shown in the equations passed the *t*-test of**
**significance at 0.05 level.**

| Season | Stepwise MLR equation | Adjusted coefficients of determination ($R^2$_adj) |
|---|---|---|
| DJF | MDA8 $O_3$ = -0.807 + 0.050*SW + 0.596*T2max + 0.016*PBLH + 0.247*PREC + 0.111*V850 + 0.066*SLP + 0.124*WS850 – 0.058*RH | 0.76 |
| MAM | MDA8 $O_3$ = -0.599 + 0.034*SW + 0.845*T2max + 0.324*V850 + 0.011*PBLH – 0.111*RH – 0.138*U850 | 0.74 |
| JJA | MDA8 $O_3$ = 0.451 + 0.067*SW + 0.530*T2max + 0.552*V850 – 0.219*RH – 0.739*WS850 + 0.012*PBLH – 0.122*SLP | 0.58 |
| SON | MDA8 $O_3$ = -1.183 – 0.076*RH + 1.303*T2max + 0.035*PBLH – 0.370*WS850 + 0.151*V850 – 0.134*PREC + 0.066*SLP | 0.76 |





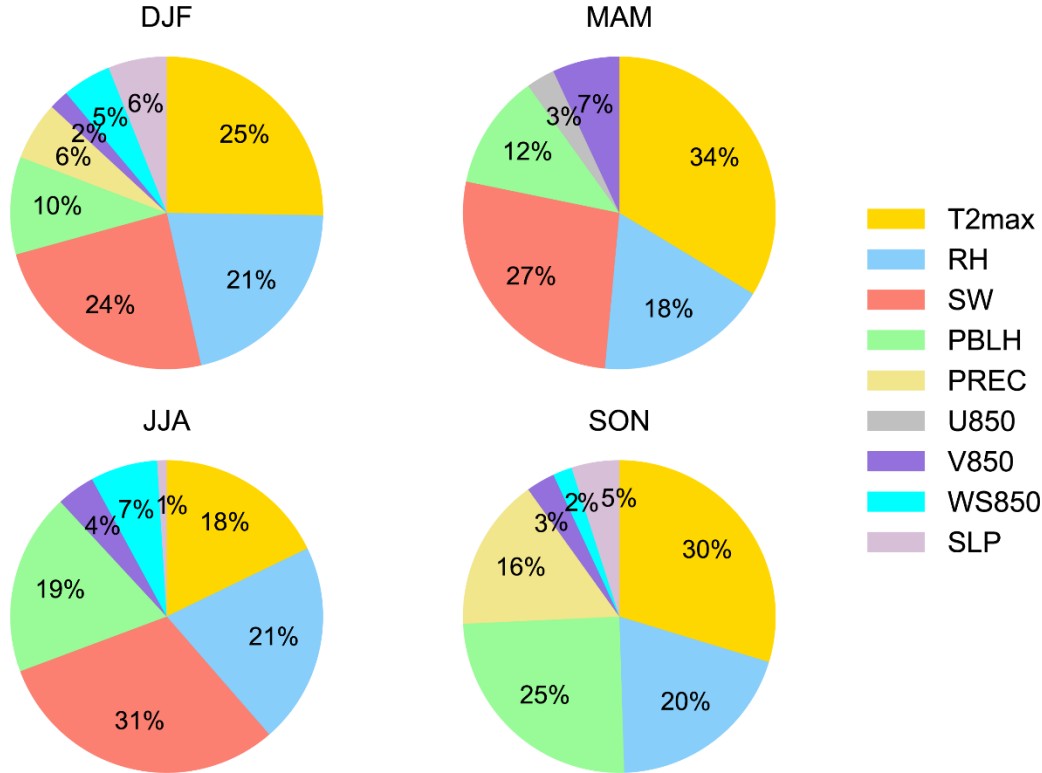

**Figure 6. The LMG decomposed contribution (%) of each meteorological variable screened by stepwise MLR model in fitting climate-driven MDA8 O$_3$ changes over EC. See Table 3 for the meanings of the abbreviations of meteorological variables.**

Large-scale regional average could obscure local characteristics, so we further conducted MLR and LMG analysis on each grid cell to identify the first and second most important meteorological parameters (hereafter called "1$^{st}$ MET" and "2$^{nd}$ MET") in China as shown in Fig. 7. In DJF, the 1$^{st}$ MET is T2max in southern EC and is SW or PBLH in northern EC, which has the relative contributions of 30-70% from LMG analyses. In JJA, the 1$^{st}$ MET is T2max in most parts of northern EC (north of 36°N), SW in most parts of central EC (26-36°N), Beijing, and Tianjin, and RH and WS850 in southern EC (south of 26°N). In the corresponding areas, T2max and SW have relative contributions of 30-70% and RH has relative contributions of 10-30%. The regional heterogeneity of the 2$^{nd}$ MET increases compared to the 1$^{st}$ MET. In DJF, the 2$^{nd}$ MET is RH in northern EC and SW in southern EC, with relative contributions of 10-30%. In JJA, the 2$^{nd}$ MET is mainly SW or T2max in northern EC and RH or WS850 in southern EC. The relative contribution of 2$^{nd}$ MET (SW or T2max) in central EC can have relative contributions of 30-50% in JJA. In summary, the key meteorological parameters for climate-induced MDA8 O$_3$ changes are not only temperature, but also SW, RH, and PBLH, depending on locations and seasons.



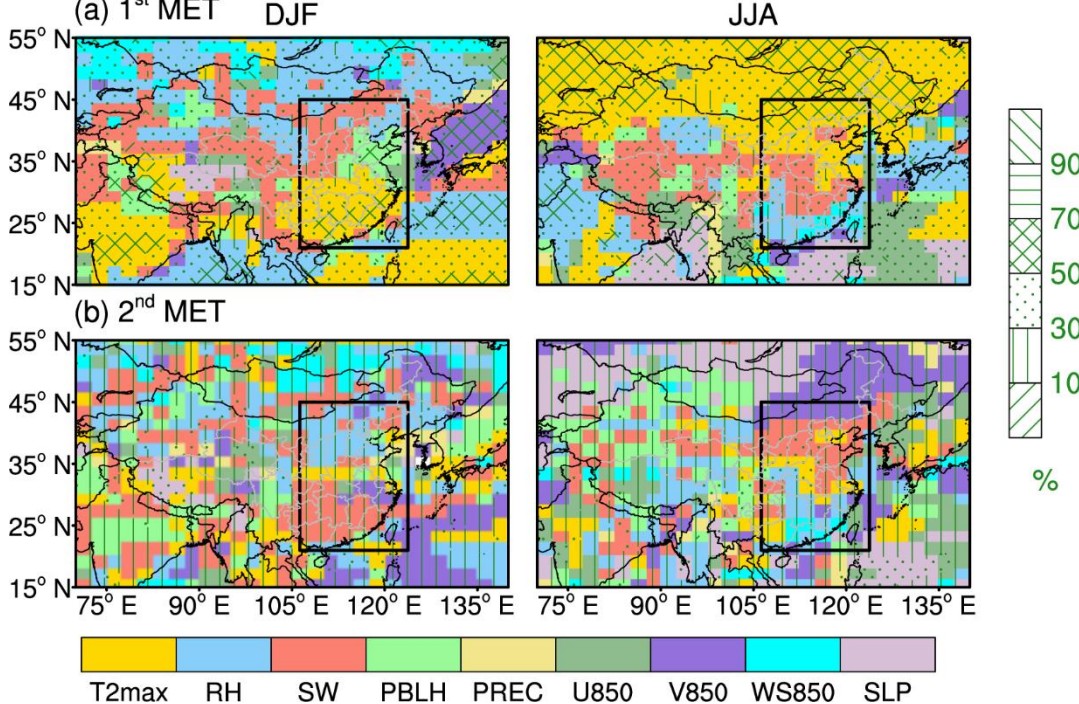

**Figure 7. The (a) 1st and (b) 2nd important meteorological parameters (1st MET and 2nd MET, respectively) for climate-induced MDA8 $O_3$ changes in China and their relative contributions in DJF and JJA., All 1st MET and 2nd MET in each 2° × 2.5° grid cell are statistically significantly correlated with MDA8 $O_3$ (p < 0.05). The overlaid fill patterns represent the relative contribution of the meteorological variable at this grid.**

**3.3.2 Key processes for climate-induced $O_3$ changes**

We performed IPR analysis to understand the intrinsic mechanism of the impact of climate change on $O_3$ in EC. Figure 8 show the vertical profiles of present-day seasonal mean $O_3$ mass and climate-driven $O_3$ mass changes of five processes (net chemical production, PBL mixing, dry deposition, cloud convection, and horizontal and vertical advection transport) in EC. Since surface $O_3$ concentrations are determined by the processes within the boundary layer (Gong and Liao, 2019), we also listed in Table 5 the present-day $O_3$ budget of five processes in EC within the boundary layer and the climate-driven $O_3$ budget changes by each process.

In present-day (Fig. 8a), net chemical production is negative at the surface due to the $O_3$ titration effect by abundant $NO_x$ and is positive in the upper levels due to the decreases in $NO_x$ concentrations and the strong solar radiation (Gong and Liao, 2019). PBL mixing refers to $O_3$ mass fluxes by turbulence within the boundary layer, which transports $O_3$ based on the concentration gradient. Since $O_3$ concentrations are higher in the upper boundary layers than at the surface (Fig. S3), PBL mixing leads to the decreases in $O_3$ in upper layers (950 to 800 hPa) and increases in surface-layer $O_3$ levels. Dry deposition occurs only at the surface, with the values of -122.1 to -37.5 Gg $d^{-1}$ in different seasons. Cloud convection process in GEOS-




Chem model describes the redistribution of species concentrations due to upward convection inside the cumulus and
subsidence outside the cumulus. Cloud convection has a large positive value below 950 hPa in all seasons due to the frequent
non-precipitation shallow convection in GISS-E2.1 (Wu et al., 2007; Miller et al., 2021) and higher $O_3$ concentrations above
950 hPa. Horizontal and vertical advection below 850 hPa is positive in DJF and negative in other seasons. For the present-
day $O_3$ budget within the boundary layer (Table 5, $PC_{CpdEpd}$), net chemical production is the dominant process that contributes
to $O_3$ budget in JJA, MAM, and SON, with the values of 136.3, 56.5, 37.6 Gg d$^{-1}$, respectively. Cloud convection has
contributions of 11.0-34.4 Gg d$^{-1}$ to $O_3$ budget. The horizontal and vertical advection is 0.4 Gg d$^{-1}$ in DJF and -23.8 to -2.7 Gg
d$^{-1}$ in other seasons.

380       Under the impact of climate change (Fig. 8b), net chemical production exhibits distinct increases below 850 hPa in all

seasons, especially in MAM and JJA. Increases in T2max and SW (Figs. 1a and c) result in increases in BVOC emission rates
by 0.4-2.9 10$^{-11}$ kg m$^{-2}$ s$^{-1}$ (Fig. S3) and in photochemical reaction rates, while decreases in RH (Fig. 1b) result in decreases in
$O_3$ destruction (Gong and Liao, 2019), which together promote the net chemical production of $O_3$. Increase in surface $O_3$ mass
by PBL mixing indicates that more $O_3$ enters the boundary layer and mixes to the surface as a result of increased PBLH (Fig.
1d). The importance of chemical process and PBL mixing corresponds well with the 1$^{st}$ and 2$^{nd}$ MET shown in Fig. 7. Dry
deposition removes more $O_3$ due to the increases in net chemical production of $O_3$. Cloud convection increases near-surface
$O_3$ mass in DJF and MAM but decreases those in JJA. Changes in horizontal and vertical advection reduce $O_3$ mass in EC at
layers below 850 hPa. Anomalous low pressure over EC in DJF indicates the presence of anomalous upward advection (Fig.
1f). Anomalous northwesterlies over northern China in other seasons obstruct the northward transport of BVOCs from southern
China and promote the outflow of $O_3$ and its precursors from EC. Circulation changes have an important effect on JJA $O_3$
concentrations, which are also confirmed by the 1st and 2$^{nd}$ MET (RH or WS850) in southern EC (Fig. 7).

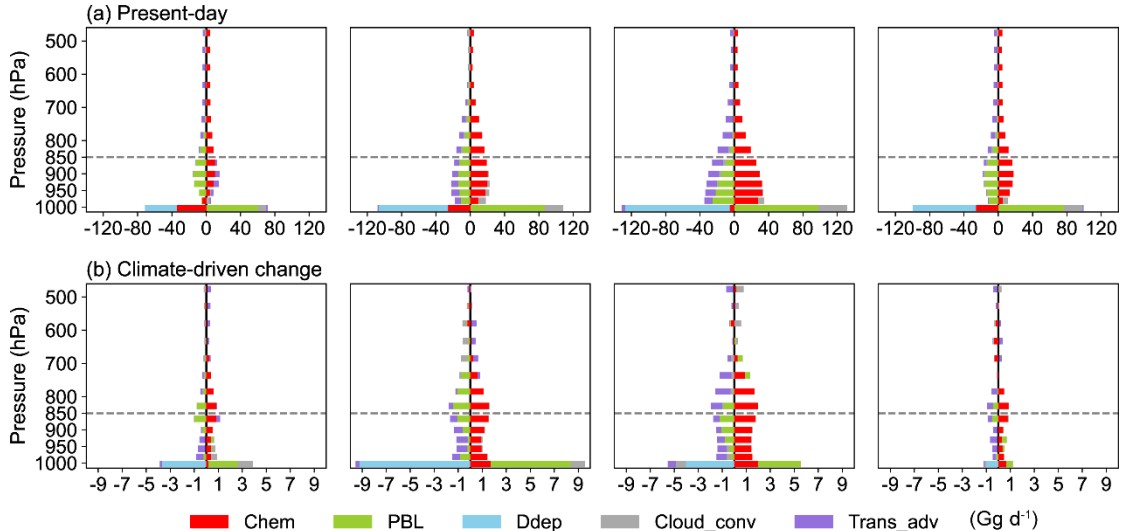


**Figure 8. (a) Vertical profile of seasonal mean $O_3$ mass (Gg d$^{-1}$) by five processes (bottom axis: net chemical production (Chem),**
**PBL mixing (PBL), dry deposition (Ddep), cloud convection (Cloud_conv), and horizontal and vertical advection (Trans_adv)) over**





EC in present-day (CpdEpd), and (b) the climate-driven changes in seasonal mean O₃ mass of each process (CfutEpd minus CpdEpd).
**All the panels have the same vertical axis in hPa.**
The sums of the climate-driven O₃ mass changes by all processes in EC are 0.6, 2.5, 6.5, and 1.7 Gg d⁻¹ in DJF, MAM,
JJA, and SON, respectively (Table 5, $PC_{DIFF}$), which are consistent with the seasonal variations in climate-induced MDA8 O₃
(Fig. 4). The net chemical production, dry deposition, and horizontal and vertical advection change by 3.3 to 16.4, -9.3 to -1.0,
and -4.3 to -0.8 Gg d⁻¹ in different seasons, respectively. The cloud convection increases by 1.5 Gg d⁻¹ in DJF and MAM and
decrease by 1.0 Gg d⁻¹ in JJA. Considering the relative contributions of individual processes (Table5, %$PC_{DIFF}$), net chemical
production is the most important process contributing to the increases of O₃ mass in all seasons, with the relative contribution
of 34.0-62.5%. Horizontal and vertical advection in JJA (-16.6%) or dry deposition in other seasons (-37.9% to -13.7%) is the
major process that reduces O₃ mass as the O₃ mass increases from chemical reactions.
**Table 5. Seasonal mean O₃ budgets (Gg d⁻¹) within the boundary layer over EC in CpdEpd ($PC_{CpdEpd}$) and CfutEpd ($PC_{CfutEpd}$).**
**The climate-driven O₃ budget changes of five process ($PC_{DIFF}$, ), and the relative contribution of each process to the total O₃ mass**
**changes (%$PC_{DIFF}$, %) are also listed, following Eqs. (2) and (3) described in Sect. 2.3.2.**

| Season | | Chemistry | PBL mixing | Dry deposition | Cloud convection | Advection transport | Total |
|--------|--------|-----------|------------|----------------|------------------|---------------------|-------|
| DJF | $PC_{CpdEpd}$ | -12.02 | 47.58 | -37.46 | 11.01 | 0.39 | 9.50 |
| | $PC_{CfutEpd}$ | -8.74 | 47.93 | -41.11 | 12.52 | -0.46 | 10.13 |
| | $PC_{DIFF}$ | 3.28 | 0.34 | -3.65 | 1.51 | -0.85 | 0.64 |
| | %$PC_{DIFF}$ | 34.04 | 3.56 | -37.88 | 15.71 | -8.80 | / |
| MAM | $PC_{CpdEpd}$ | 56.48 | 50.39 | -80.71 | 25.83 | -11.43 | 40.56 |
| | $PC_{CfutEpd}$ | 68.13 | 50.84 | -89.96 | 27.37 | -13.35 | 43.03 |
| | $PC_{DIFF}$ | 11.65 | 0.45 | -9.25 | 1.54 | -1.92 | 2.47 |
| | %$PC_{DIFF}$ | 46.95 | 1.81 | -37.28 | 6.21 | -7.75 | / |
| JJA | $PC_{CpdEpd}$ | 136.26 | 35.23 | -122.07 | 34.37 | -23.78 | 60.01 |
| | $PC_{CfutEpd}$ | 152.61 | 34.75 | -126.09 | 33.41 | -28.13 | 66.55 |
| | $PC_{DIFF}$ | 16.35 | -0.48 | -4.03 | -0.96 | -4.34 | 6.54 |
| | %$PC_{DIFF}$ | 62.49 | -1.84 | -15.39 | -3.67 | -16.59 | / |
| SON | $PC_{CpdEpd}$ | 37.58 | 41.58 | -73.96 | 22.75 | -2.71 | 25.23 |
| | $PC_{CfutEpd}$ | 41.99 | 40.61 | -74.95 | 22.82 | -3.50 | 26.97 |
| | $PC_{DIFF}$ | 4.42 | -0.97 | -0.99 | 0.07 | -0.79 | 1.74 |
| | %$PC_{DIFF}$ | 61.02 | -13.45 | -13.65 | 0.97 | -10.90 | / |





**3.4 Projections of climate-driven MDA8 O₃ changes from the CMIP6 models**

In Sect. 5.1, we applied the stepwise MLR model to relate 10 yr daily MDA8 O₃ anomalies to 10 yr daily meteorological parameter anomalies at each grid cell and obtained the corresponding MLR equation. The climate-driven seasonal mean MDA8 O₃ concentration changes projected by stepwise MLR model at each grid cell can be obtained by substituting the corresponding seasonal mean meteorological parameter anomalies of GCAP 2.0 into the regression equations obtained by daily anomalies above, which will be referred to as Dev_MLR_MDA8 hereafter. The Dev_MLR_MDA8 values for a target region are then obtained by averaging over all the grid cells in the region. We selected EC, NCP, and YRD as the target regions in this study. Figures 9a-c evaluate the seasonal and annual mean Dev_MLR_MDA8 values averaged over EC, NCP, and YRD by comparing them with the simulated values by GCAP 2.0 (hereafter called Dev_GCAP2_MDA8). The seasonal and annual mean values of Dev_MLR_MDA8 and Dev_GCAP2_MDA8 are exactly the same, with the R value of 1.0 and the NMB value of 0.0% in all three regions. In China, the spatial distributions and magnitudes of the seasonal mean Dev_MLR_MDA8 values are consistent with the seasonal mean Dev_GCAP2_MDA8 values (Fig. S4), with high pattern correlation coefficients of 1.0 in four seasons, indicating that it is feasible to predict climate-driven MDA8 O₃ concentration changes by stepwise MLR model. Therefore, we input the corresponding seasonal mean meteorological parameter anomalies from the six CMIP6 models into the regression equations to obtain multi-model projections of climate-induced MDA8 O₃ changes under carbon neutrality scenario.

Figures 9d-f shows the climate-driven seasonal and annual mean MDA8 O₃ changes averaged over EC, NCP, and YRD regions predicted by stepwise MLR model using meteorology anomalies from the GCAP 2.0 and other six CMIP6 models under SSP1-1.9 scenario. The Dev_MLR_MDA8 values of GCAP 2.0 and all six CMIP6 models are positive throughout the year in all three regions, indicating that climate change will increase MDA8 O₃ concentrations over polluted regions in China even under carbon neutrality scenario. Similar to the GCAP 2.0 results, the Dev_MLR_MDA8 values of all six CMIP6 models in the three regions are much larger in JJA than in other seasons, with the values in the range of 2.9-4.2, 6.5-9.4, and 3.3-8.5 ppbv in EC, NCP, and YRD, respectively. In JJA, the Dev_MLR_MDA8 values of MMM (average of six CMIP6 models) are 3.5, 7.5, and 5.1 ppbv in EC, NCP, and YRD, respectively, higher than the Dev_MLR_MDA8 values of GCAP 2.0 of 2.3, 4.7, and 3.0 pbbv, respectively. In other seasons, the Dev_MLR_MDA8 values of MMM are in the range of 0.9-1.4, 1.2-2.3, and 1.2-2.2 ppbv in EC, NCP, and YRD, respectively, and the Dev_MLR_MDA8 values of GCAP 2.0 are in the range of 0.4-1.3, 0.4-2.8, and 0.3-1.7 pbbv, respectively. Overall, the Dev_MLR_MDA8 values of GCAP 2.0 tend to be in the lower end of the multi-model projection results, especially in JJA.



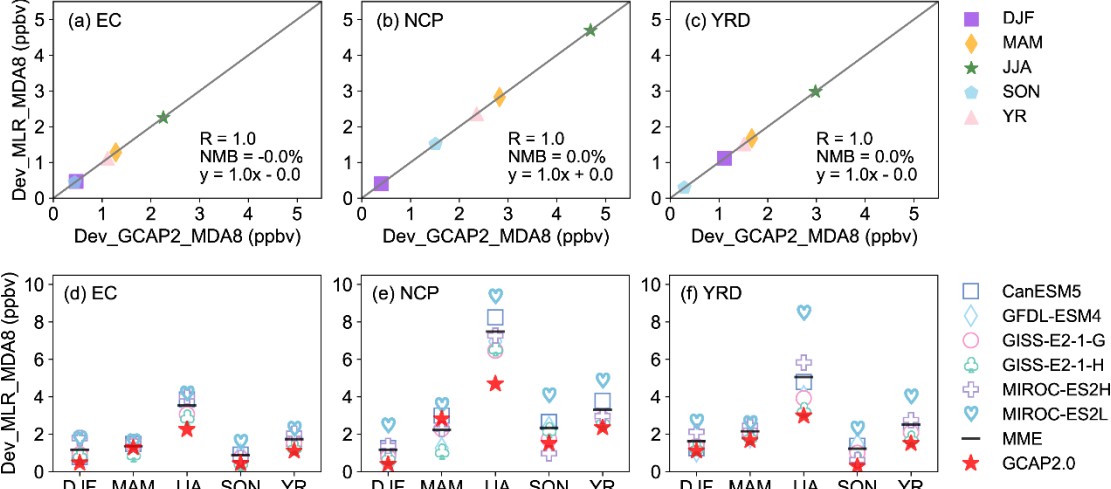

**Figure 9. (a)-(c) The scatterplot of climate-induced MDA8 O₃ changes (ppbv) simulated by GCAP 2.0 (Dev_GCAP2_MDA8) versus those projected by MLR model (Dev_MLR_MDA8) in EC, NCP, and YRD regions. The correlation coefficient (R), normalized mean biases (NMB), and linear fit (grey solid line and equation) are also shown. (d)-(f) The climate-driven seasonal and annual mean MDA8 O₃ concentration changes (ppbv) projected by MLR model using the climate outputs from GCAP 2.0 and six CMIP6 models under SSP1-1.9 scenario. The multi-model mean (MMM) is calculated from the average of the six CMIP6 models. Different markers represent different models, black lines represent MMM, and red stars represent GCAP 2.0 results.**

### 3.5 Future changes in tropospheric O₃ driven by changes in anthropogenic emissions

We show large impact of climate change on tropospheric O₃ in previous sections, so it is of interest to examine briefly the effects of emission changes on surface O₃ levels (CpdEfut minus CpdEpd) under carbon neutrality scenario as shown in Fig. 4b. Emission change alone leads to decreases in MDA8 O₃ concentrations of 0.5 (1.6%), 8.0 (16.7%), 15.8 (27.1%), and 7.0 ppbv (16.5%) over EC in DJF, MAM, JJA, and SON, respectively. Although the regional mean MDA8 O₃ concentrations in EC decrease in all seasons, the nationwide decreases in MDA8 O₃ concentration occur only in JJA. In other seasons, MDA8 O₃ concentrations in northern China increase owing to changes in anthropogenic emissions, with the maximum increases of 8-12 ppbv in DJF. The regional mean MDA8 O₃ concentrations in NCP increase by 6.7 (34.3%) in DJF and 2.2 ppbv (6.7%) in SON, and those in YRD increase by 3.0 ppbv (9.5%) in DJF.

The increases in MDA8 O₃ concentrations by changes in anthropogenic emissions under carbon neutrality scenario can be explained by O₃ formation regime. Figure 10 shows the present-day seasonal mean formaldehyde nitrogen ratio (FNR), which was introduced by Jin and Holloway (2015) to show O₃ sensitivity to its precursors (see S1 in Supplementary Material). In DJF, FNR values in eastern China are lower than 1, indicating a general VOC-limited regime. In MAM and SON, the VOC-limited regime shrinks toward the North China, and South China is in the NOₓ-limited (FNR values exceeding 2) or transitional (FNR values between 1 and 2) regime. In JJA, most of China is in the NOₓ-limited regime, while the NCP region is still in the VOC-limited or transitional regime. Although the anthropogenic emissions of VOCs and NOₓ in NCP decrease largely (70-



90%) under SSP1-1.9 scenario (Fig. S5), MDA8 $O_3$ concentrations in this region increase in the future in DJF, MAM, and
SON because NCP is in the VOC-limited regime.
Overall, considering the combined effects of climate change and emission change (CfutEfut minus CpdEpd), the spatial
distributions and magnitudes of MDA8 $O_3$ changes are similar to those considering the emission changes alone (Fig. 4c),
indicating that future MDA8 $O_3$ concentrations are dominated by emission changes. However, the effects of climate penalty
(0.5-2.3, 0.4-4.7, and 0.3-3.0 ppbv in EC, NCP, and YRD, respectively) cannot be ignored. Note that changes in both climate
and emissions lead to increases in MDA8 $O_3$ in DJF and SON over NCP and in DJF over YRD, calling for more attention to
these regions in future $O_3$ pollution control strategies.

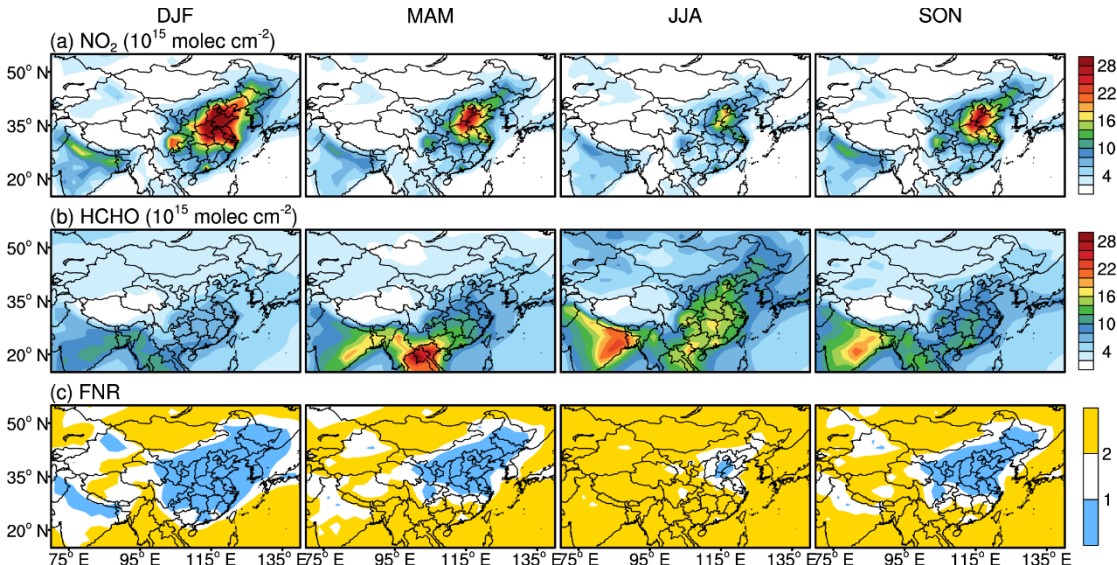

**Figure 10. Distributions of seasonal mean tropospheric columns of (a) nitrogen dioxide (NO$_2$) and (b) formaldehyde (HCHO) ($10^{15}$**
**molec cm$^{-2}$), and (c) formaldehyde nitrogen ratio (FNR) in present-day.**

## 4 Conclusions

In this study, we quantify the effects of climate changes over 2010-2045 on $O_3$ levels in China under carbon neutrality
scenario (SSP1-1.9 scenario), focusing on the key meteorological parameters and processes for understanding the climate-
induced $O_3$ changes by using the GCAP 2.0, stepwise MLR model, LMG method, and IPR analysis. The uncertainties in future
$O_3$ levels resulted from the uncertainties in simulated future climate are also quantified by using outputs of climate from CMIP6
models.
Under carbon neutrality scenario, over EC, GCAP 2.0 and all six CMIP6 models project the increases in T2max, SW,
and PBLH in all seasons, and most models project reductions in RH and SLP and increases in PREC. Projected annual mean
changes over EC in T2max, SW, PBLH, PREC, RH, and SLP are, respectively, 1.4 K, 11.8 W m$^{-2}$, 30.6 m, 0.3 mm day$^{-1}$, -



0.7%, and -0.3 hPa on a multi-model mean (MMM) basis and 1.1 K, 7.3 W m$^{-2}$, 23.7 m, 0.03 mm day$^{-1}$, -1.3%, and -0.3 hPa
from GCAP 2.0. Relative to the MMM, GCAP 2.0 underestimates the increases in T2max, SW, PBLH, and PREC and
overestimates the decreases in RH.
The GCAP 2.0 model generally reproduces the spatial distribution and magnitude of observed seasonal mean MDA8 $O_3$
concentrations, with R values of 0.12-0.63 and NMB values of 7.1-18.6% in different seasons. Climate change over 2010-
2045 under the carbon neutrality scenario is simulated by GCAP 2.0 to increase the regional mean MDA8 $O_3$ concentrations
by 0.4-2.3 ppbv (1.0-3.9%) over EC, 0.4-4.7 ppbv (2.0-7.3%) over NCP, and 0.3-3.0 ppbv (0.6-5.1%) over YRD in different
seasons, with the maximum increases in JJA. By using the stepwise MLR model, we find that changes in the key meteorological
variables retained in MLR equations can explain 58-76% of the climate-driven MDA8 $O_3$ concentration changes over EC. By
using the LMG method, we find that the most important meteorological parameters for climate-induced MDA8 $O_3$ changes
are not only temperature, but also SW, RH, and PBLH, depending on locations and seasons. Corresponding to these changes
in meteorological parameters, IPR analysis shows that net chemical production (accounting for 34.0-62.5% of total $O_3$ mass
change caused by all processes within the boundary layer) is the most important process contributing to the climate-induced
increases of $O_3$ mass in all seasons. Horizontal and vertical advection in JJA (-16.6%) or dry deposition in other seasons (-
37.9% to -13.7%) is the major process that reduces $O_3$ mass.
Under carbon neutrality scenario, future MDA8 $O_3$ concentration changes in EC are dominated by changes in
anthropogenic emissions (decrease by 0.5-15.8 ppbv), however, the effects of climate penalty (increase by 0.5-2.3 ppbv from
GCAP 2.0) cannot be ignored. Both climate changes and emission changes increase MDA8 $O_3$ values in DJF and SON over
NCP and in DJF over YRD, indicating that these regions require more attention in future $O_3$ pollution control.
The estimate of the effect of climate change on $O_3$ pollution by using a single model GCAP 2.0 may have uncertainties.
Therefore, we also obtain the multi-model projection results of future MDA8 $O_3$ changes driven by 2010-2045 climate change
under carbon neutrality scenario by using stepwise MLR model. In JJA, six CMIP6 models project increases in MDA8 $O_3$
ranging from 2.9-4.2, 6.5-9.4, and 3.3-8.5 ppbv in EC, NCP, and YRD, respectively, indicating that GCAP 2.0 results (2.3,
4.7, and 3.0 pbbv) are in the lower end of the multi-model projections.
**Data availability**
The observed hourly surface $O_3$ concentrations in 2015 are derived from the China National Environmental Monitoring Center
(https://air.cnemc.cn:18007/, CNEMC). The satellite observations of $NO_2$ and HCHO are downloaded from
https://www.temis.nl/airpollution/. The climate outputs from GCAP 2.0 and other six CMIP6 models can be downloaded from
http://atmos.earth.rochester.edu/input/gc/ExtData/GCAP2/CMIP6/ and https://esgf-node.llnl.gov/search/cmip6/, respectively.
The GEOS-Chem model is available at http://wiki.seas.harvard.edu/geos-chem/index.php/GEOS-Chem_13.2.1. The
anthropogenic and biomass burning emission inventory of SSP1-1.9 are available from



https://aims2.llnl.gov/search/input4mips/. The simulation results are available upon request from the corresponding author
(hongliao@nuist.edu.cn).

**Author contributions**

LK and HL conceived the study and designed the experiments. LK carried out the model simulations and performed the data
analysis. KL, XY, YY, and YW provided useful comments on the paper. LK and HL prepared the paper.

**Competing interests**

The authors declare that they have no conflict of interest.

**Acknowledgements**

We acknowledge the CNEMC, Tropospheric Emission Monitoring Internet Service (TEMIS), and CMIP6 teams for making
their data publicly available. We acknowledge the efforts of GEOS-Chem working groups for developing and managing the
model.

**Financial support**

This work was supported by the National Natural Science Foundation of China under grants 42293320 and 42021004.

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
