# Peer review of "Effects of 2010-2045 climate change on ozone levels in China under 1 carbon neutrality scenario: Key meteorological parameters and 2 processes 3"

_EGUsphere, 2024_

## Referee Comment (RC1)

**Review of Kang et al., 2024 (ACPD)**

Title: *Effects of 2010-2045 climate change on ozone levels in China under carbon neutrality scenario: Key meteorological parameters and processes*

The overall manuscript is well-documented, but I have some major concerns and suggestions for improvement:

1. The title emphasizes the effects of climate change but does not highlight emissions, which have a much higher impact on ozone levels compared to climate change. Since the manuscript examines both, the title should reflect the role of emissions more explicitly.
2. GCAP2.0 is a one-way offline model, and the meteorology you used to drive GEOS-Chem is parameterized. In the "Results" section (e.g., Figure 1), meteorological variables are shown. Are these variables inputs or outputs of the model? Please clarify. Furthermore, it is crucial to clearly define what is considered a climate variable in this study and describe the differences in these variables between present-day and future scenarios, similar to the approach used for emissions (Section 2.2.2). Additionally, since GCAP2.0 is a one-way offline model, do changes in emissions have any feedback effect on meteorology? I assume not, but this should be explicitly addressed.
3. The manuscript frequently discusses regions like EC, NCP, or YRD. Instead of presenting results for all of China, zooming in on these regions while plotting would provide more clarity, particularly for localized changes.
4. The model's performance in capturing present-day results (e.g., Figure 3) is concerning. For instance, the MAM R-value is only 0.12, indicating a poor representation of trends. This raises questions about the reliability of future projections. Moreover, your results are at the lower end of CMIP6 model projections. Please provide a detailed explanation of why GCAP behaves differently, even for regional means.
5. For difference plots of the same variable, use a consistent color scale to facilitate comparison of magnitudes across different forcing factors. For example, in Figure 4, ensure the scales for "climate," "emissions," and "combined" effects are the same.
6. "Climate + Emissions" represents the combined effect of both forcings. Have you tried linearly summing the individual effects of climate and emissions and comparing this sum to the combined effect? If not, this analysis should be performed and discussed.
7. BVOC emissions are included in the "emissions" forcing. Since MEGAN is used, "climate" forcing also influences BVOCs. This raises the possibility of double-counting BVOC emissions in the combined effect. If double-counting is not an issue, please clarify this in the manuscript.
8. The manuscript states that meteorology explains 58–76% of the total change, yet net chemical production is described as the most important process. This appears contradictory. Please reconcile and clearly quantify the contributions of meteorological and chemical factors to the total change.
9. The manuscript omits some recent global studies on the climate effect on ozone, such as Bhattarai et al. (2024) (STOTEN; https://doi.org/10.1016/j.scitotenv.2023.167759). Discussing your findings in the context of these studies would strengthen the manuscript.

10. Figure 1: Clearly indicate what the difference plots represent in the caption and text. For example, is the change shown as CfutEfut − CpdEpd, or is it only the effect of climate? The figure caption should be self-explanatory.
11. Line 409: There is no section called 5.1
12. Consider adding a discussion on the policy-relevant implications of the carbon neutrality scenario towards the end of the manuscript.
13. The carbon neutrality target is 2060, but you selected 2045 as the endpoint of your analysis. Is there any reason behind this?

---

## Author Comment (AC1)

**Response to Comments of Reviewer #1**

**Manuscript number:** acp-2024-3470
**Title:** Effects of 2010-2045 climate change on ozone levels in China under carbon neutrality scenario: Key meteorological parameters and processes

**General comments:** The overall manuscript is well-documented, but I have some major concerns and suggestions for improvement:

Thanks to the referee for the helpful comments and constructive suggestions. We have revised the manuscript carefully and the point-to-point responses are listed below.

**Major concerns/questions:**

1. The title emphasizes the effects of climate change but does not highlight emissions, which have a much higher impact on ozone levels compared to climate change. Since the manuscript examines both, the title should reflect the role of emissions more explicitly.

**Response:**

This manuscript is focused on the effects of climate change on $O_3$ levels, with detailed analyses on key meteorological parameters and processes to understand the climate-induced $O_3$ changes. Although the effects of changes in anthropogenic emissions on $O_3$ levels are briefly discussed in Section 3.5, these discussions aim only to provide a reference level for the understanding of the magnitude of climate-induced changes in $O_3$.

As shown in the third paragraph of "Introduction", several papers have already quantified the effects of changes in anthropogenic emissions on $O_3$ under carbon neutrality scenario (Shi et al., 2021; Xu et al., 2022; Wang et al., 2023). However, none of them examined the effects of future climate change. We feel that the current title can better highlight the novelty of our work.

2. GCAP2.0 is a one-way offline model, and the meteorology you used to drive GEOS-Chem is parameterized. In the "Results" section (e.g., Figure 1), meteorological variables are shown. Are these variables inputs or outputs of the model? Please clarify. Furthermore, it is crucial to clearly define what is considered a climate variable in this study and describe the differences in these variables between present-day and future scenarios, similar to the approach used for emissions (Section 2.2.2). Additionally, since GCAP2.0 is a one-way offline model, do changes in emissions have any feedback effect on meteorology? I assume not, but this should be explicitly addressed.

**Response:**

GCAP 2.0 model framework, developed by Murray et al. (2021), is a one-way offline coupling between the GISS-E2.1 GCM and the GEOS-Chem model. Meteorology for driving GEOS-Chem model (namely GCAP 2.0 meteorology) is archived from the climate outputs of GISS-E2.1 GCM. Therefore, meteorological variables shown in Figure 1 are outputs of the GISS-E2.1 GCM model.

In this work, we use the 10-yr average of GCAP 2.0 meteorology to represent climatology. The GCAP 2.0 meteorology averaged over 2005-2014 is used to represent the present-day climate (2010), and that averaged over 2040-2049 under SSP1-1.9 scenario is used to represent the future climate (2045). All meteorological variables shown in Figure 1 are climate variables, and their differences between present-day and future (under SSP1-1.9 scenario) are presented in Figure 1 and described in detail in Section 3.1.1.

As discussed above, GCAP 2.0 is a one-way offline coupling between the GISS-E2.1 GCM and the GEOS-Chem model, so changes in emissions have no feedback effect on meteorology.

3. The manuscript frequently discusses regions like EC, NCP, or YRD. Instead of presenting results for all of China, zooming in on these regions while plotting would provide more clarity, particularly for localized changes.

**Response:**

Following your suggestions, we carefully examined our presentation in the text and all figures, and found that two figures (Figures 2 and 9) are presented in terms of regions such as EC, NCP, or YRD. Figure 2 shows the 2010-2045 changes in seasonal mean meteorological parameters in EC, and their corresponding localized changes can be seen in Figure 1. Figure 9 shows the climate-driven seasonal and annual mean MDA8 $O_3$ concentration changes projected by MLR model using the climate outputs from GCAP 2.0 and six CMIP6 models under SSP1-1.9 scenario. To see localized changes, we have added Figure S5 (see below) in the Supplementary Material to see the spatial characteristics corresponding to Figure 9. We have also added the following sentences in the second paragraph of Section 3.4 to describe the localized changes: "The spatial distributions of climate-driven changes in annual mean MDA8 $O_3$ concentrations from GCAP 2.0 and the other six CMIP6 models are shown in Fig. S5. The climate-induced increases in annual mean MDA8 $O_3$ predicted by all models are mainly concentrated in central and northern EC. In NCP and its surrounding areas, while the maximum increases in annual mean MDA8 $O_3$ concentrations were simulated to be 2-4 ppbv from GCAP 2.0, the values were 4-8 ppbv from four of the six CMIP6 models.".

[Figure]

Figure S5. The spatial distributions of climate-driven changes in annual mean MDA8 $O_3$ concentrations (ppbv) in EC projected by MLR model using the climate outputs from GCAP 2.0 and the other six CMIP6 models under SSP1-1.9 scenario. The multi-model mean (MMM) is calculated from the average of the six CMIP6 models.

4.  The model's performance in capturing present-day results (e.g., Figure 3) is concerning. For instance, the MAM R-value is only 0.12, indicating a poor representation of trends. This raises questions about the reliability of future projections. Moreover, your results are at the lower end of CMIP6 model projections. Please provide a detailed explanation of why GCAP behaves differently, even for regional means.

**Response:**

In MAM, compared with the observations, the GEOS-Chem model has low biases in NCP and YRD (with the NMBs of -24.0% and -6.7%, respectively) but high biases outside these two regions (with a NMB of 9.7%), leading to a low spatial correlation coefficient (R) of 0.12 over the whole of China. The fairly low biases in NCP and YRD in the GEOS-Chem model are not expected to affect our projections of the future changes in MDA8 $O_3$ concentrations.

Our results are at the lower end of CMIP6 model projections, which can be explained by the differences in key meteorological parameter anomalies between GCAP 2.0 and six CMIP6 models.

For example, as shown in Section 3.3.1, the top two most important meteorological variables over EC in DJF are T2max and SW (Figures 6 and 7), and their corresponding changes over 2010-2045 projected by GCAP 2.0 are 1.3 K and 4.4 W m$^{-2}$ (Figure 2). However, the projected future changes in T2max and SW over 2010-2045 from CMIP6 multi-models are in the range of 1.0-2.1 K and 8.7-11.2 W m$^{-2}$. Therefore, the underestimation of the increases in T2max and SW in GCAP 2.0 leads to the underestimation of net chemical productions of O$_3$. As a result, the increases in MDA8 O$_3$ predicted by GCAP 2.0 are at the lower end of CMIP6 multi-model projections.

5.  For difference plots of the same variable, use a consistent color scale to facilitate comparison of magnitudes across different forcing factors. For example, in Figure 4, ensure the scales for "climate," "emissions," and "combined" effects are the same.

**Response:**
Thanks for the suggestion. We have revised Figure 4 as suggested (see below).

[Figure]

Figure 4. Predicted future changes in seasonal mean MDA8 O$_3$ concentrations (ppbv) due to (a) climate change alone (CfutEpd minus CpdEpd), (b) emission change alone (CpdEfut minus CpdEpd), and (c) combined climate and emission changes (CfutEfut minus CpdEpd) under SSP1-1.9 scenario. The black, green and blue rectangles indicate the domain of EC, NCP, and YRD, respectively. The dotted areas represent a statistically significant difference at the 95% level according to Student's two sample t test. The values at the top right of each panel are the regional mean values of EC, NCP, and YRD, respectively.

6.  "Climate + Emissions" represents the combined effect of both forcings. Have you tried linearly summing the individual effects of climate and emissions and comparing this sum to the combined effect? If not, this analysis should be performed and discussed.

**Response:**

Following the Reviewer's suggestion, we have added the following sentences in the last paragraph of Section 3.5:"Note that the sum of the individual effects of climate (Fig. 4a) and emissions (Fig. 4b) is not equal to the combined effects (Fig. 4c) due to the nonlinear relationship between the simulations (Dang et al., 2021).".

7. BVOC emissions are included in the "emissions" forcing. Since MEGAN is used, "climate" forcing also influences BVOCs. This raises the possibility of double-counting BVOC emissions in the combined effect. If double-counting is not an issue, please clarify this in the manuscript.

**Response:**

As shown in Section 2.2.2, BVOC emissions are computed using MEGAN, which is driven by meteorological conditions, and thus BVOC emissions are included in the "climate" forcing. The effects of emissions only include changes in anthropogenic emissions and biomass burning emissions, so we do not double-count BVOC emissions in the combined effects. To clarify this, we have added the following sentences in the second paragraph of Section 2.2.2: "Changes in all natural emissions are calculated by using projected climate change, which are considered as the effects of climate change.".

8. The manuscript states that meteorology explains 58–76% of the total change, yet net chemical production is described as the most important process. This appears contradictory. Please reconcile and clearly quantify the contributions of meteorological and chemical factors to the total change.

**Response:**

The effects of climate change are quantified by CfutEpd minus CpdEpd. This sentence here means that the key meteorological parameters selected among all climate variables can explain 58-76% of the total effects of climate change. For example, as shown in Table 4, in JJA, changes in SW, T2max, V850, RH, WS850, PBLH, and SLP explained 58% of the climate-induced changes in MDA8 $O_3$ over EC.

The climate-driven $O_3$ changes depend on the net effect of changes in physical and chemical processes, including net chemical production, PBL mixing, dry deposition, cloud convection, and horizontal and vertical advection transport. As shown in Section 3.3.2, net chemical production has a relative contribution of 34.0-62.5% among all the five processes, hence it is the most important process.

Therefore, as discussed above, they are not contradictory.

9. The manuscript omits some recent global studies on the climate effect on ozone, such as Bhattarai et al. (2024) (STOTEN; https://doi.org/10.1016/j.scitotenv.2023.167759). Discussing your findings in the context of these studies would strengthen the manuscript.

**Response:**

Following the Reviewer's suggestion, we have added the following sentences in the first paragraph of Section 3.2.2: "Our results are lower than the recent study by Bhattarai et al. (2024), who reported that climate change alone could lead to an increase of 5-15 ppbv in JJA MDA8 $O_3$ levels in EC over 2010-2050 under SSP1-2.6 scenario by using Community Earth System Model (CESM) and Community Atmospheric Model version 4 with chemistry (CAM4-chem).".

10. Figure 1: Clearly indicate what the difference plots represent in the caption and text. For example, is the change shown as CfutEfut − CpdEpd, or is it only the effect of climate? The figure caption should be self-explanatory.

**Response:**

Figure 1 shows projected climate change by GISS-E2.1 GCM from 2010 (averaged over 2005-2014) to 2045 (averaged over 2040-2049) under SSP1-1.9 scenario (see our response to your specific

Comment #2).

11. Line 409: There is no section called 5.1

**Response:**

We have changed "In Sect. 5.1" to "In Sect. 3.3.1".

12. Consider adding a discussion on the policy-relevant implications of the carbon neutrality scenario towards the end of the manuscript.

**Response:**

Following the Reviewer's suggestion, we have added discussions on the policy-relevant implications in the last paragraph of "Conclusion" section: "Additionally, MDA8 $O_3$ concentrations increase by changes in anthropogenic emissions in the future in DJF, MAM, and SON despite the large reductions in $NO_x$ and VOCs (70-90%) in North China (Fig. S6) under SSP1-1.9 scenario, indicating an urgent need to find appropriate emission reduction ratios of VOCs and $NO_x$ based on $O_3$ sensitivity to precursors and to climate for effective future $O_3$ pollution control in China.".

13. The carbon neutrality target is 2060, but you selected 2045 as the endpoint of your analysis. Is there any reason behind this?

**Response:**

The GCAP 2.0 meteorology only contains four time slices: pre-industrial era (1851-1860), recent past (2001-2014), near-future (2040-2049), and end-of-the-century (2090-2099). The closest time slice of GCAP 2.0 meteorology to carbon neutrality target year 2060 is 2040-2049. Therefore, the year 2045 (averaged over 2040-2049) is selected as the endpoint for the analysis considering the available GCAP 2.0 meteorology. To make this clear, we have added the following sentences in the first paragraph of Section 2.2.3: "The GCAP 2.0 meteorology are available for four time slices: pre-industrial era (1851-1860), recent past (2001-2014), near-future (2040-2049), and end-of-the-century (2090-2099).".

**References:**

Bhattarai, H., Tai, A. P. K., Val Martin, M., and Yung, D. H. Y.: Impacts of changes in climate, land use, and emissions on global ozone air quality by mid-21st century following selected Shared Socioeconomic Pathways, Sci. Total Environ., 906, https://doi.org/10.1016/j.scitotenv.2023.167759, 2024.

Dang, R., Liao, H., and Fu, Y.: Quantifying the anthropogenic and meteorological influences on summertime surface ozone in China over 2012-2017, Sci. Total Environ., 754, 142394, https://doi.org/10.1016/j.scitotenv.2020.142394, 2021.

Murray, L. T., Leibensperger, E. M., Orbe, C., Mickley, L. J., and Sulprizio, M.: GCAP 2.0: a global 3-D chemical-transport model framework for past, present, and future climate scenarios, Geosci. Model Dev., 14, 5789-5823, https://doi.org/10.5194/gmd-14-5789-2021, 2021.

Shi, X., Zheng, Y., Lei, Y., Xue, W., Yan, G., Liu, X., Cai, B., Tong, D., and Wang, J.: Air quality benefits of achieving carbon neutrality in China, Sci. Total Environ., 795, https://doi.org/10.1016/j.scitotenv.2021.148784, 2021.

Wang, Y., Liao, H., Chen, H., and Chen, L.: Future Projection of Mortality From Exposure to $PM_{2.5}$ and $O_3$ Under the Carbon Neutral Pathway: Roles of Changing Emissions and Population Aging, Geophys. Res. Lett., 50, https://doi.org/10.1029/2023gl104838, 2023.

Xu, B., Wang, T., Ma, D., Song, R., Zhang, M., Gao, L., Li, S., Zhuang, B., Li, M., and Xie, M.: Impacts of regional emission reduction and global climate change on air quality and temperature to attain carbon neutrality in China, Atmos. Res., 279, https://doi.org/10.1016/j.atmosres.2022.106384, 2022.

---

## Author Comment (AC2)

**Response to Comments of Reviewer #2**

**Manuscript number:** acp-2024-3470
**Title:** Effects of 2010-2045 climate change on ozone levels in China under carbon neutrality scenario: Key meteorological parameters and processes

**General comments:** This study investigates the impact of 2010-2045 climate changes on the ozone levels under carbon neutrality scenario using the GISS-E2.1 GCM and the GEOS-Chem models. The results of this study have important implication to the future air pollution control strategy development. The paper is well written. I recommend its acceptance for publication after some minor revisions.

Thanks to the referee for the helpful comments and constructive suggestions. We have revised the manuscript carefully and the point-to-point responses are listed below.

**Major concerns/questions:**

1.  Line 52-53: It is a 33% reduction in 90th MDA8 $O_3$, rather than 84%.

**Response:**

Thanks for pointing this out. We have changed "84%" to "33%".

2.  Line 151-154: A detailed description about the SSPs inventory is suggested. For the present-day anthropogenic emissions (year 2015), the MEIC inventory is more widely used to drive air quality models in China. What is the different in various pollutant emissions between SSPs and MEIC emission inventories for the year 2015? Moreover, for the future biomass burning emission inventory, how is it developed? If wild fire emissions are included, is it considered the impact of future climate changes?

**Response:**

Description of SSPs inventory is presented in the second paragraph of "Introduction" section: "Shared Socioeconomic Pathways (SSPs) are the state-of-the-art global emission scenarios, which combines socioeconomic and technological development with future climate radiative forcing outcomes (RCPs) into a scenario matrix architecture (Gidden et al., 2019). Gidden et al. (2019) constructed nine scenarios of future emissions trajectories, including SSP1-1.9, SSP1-2.6, SSP2-4.5, SSP3-7.0, SSP3-LowNTCF, SSP4-3.4, SSP4-6.0, SSP5-3.4-Overshoot (OS), and SSP5-8.5. Among all scenarios, only the SSP1-1.9 scenario achieves net negative emissions of carbon dioxide ($CO2$) for China and the world by 2060 (Wang et al., 2023), and thus we defined it as the carbon neutrality scenario and applied in this work."

To make it more detailed, we have added the following sentences in the first paragraph of Section 2.2.2: "Global anthropogenic and biomass burning emissions of pollutants are from the SSP1-1.9 inventory, which has a monthly temporal resolution and a 0.5° spatial resolution. The anthropogenic emissions in SSPs are from nine sectors (including agricultural, energy, industry, transportation, residential and commercial, solvents production and application, waste, international shipping, and aircraft), and the biomass burning emissions are from four sectors (including agricultural waste burning, forest burning, grassland burning, and peat burning) (Gidden et al., 2019). Future anthropogenic and biomass burning emission are obtained from the integrated assessment model (IAMs) results for each SSPs scenario after harmonization (enabling consistent transitions from the historical data used in CMIP6 to future trajectories) and downscaling (improving the spatial resolution of emissions) (Gidden et al., 2019). The impacts of future climate change on biomass burning emissions (including wild fire emissions) are not considered.".

The annual total anthropogenic emissions of $NO_x$, CO, NMVOCs, $SO_2$, $NH_3$, OC, and BC over China in 2015 are 33.6, 190.0, 30.0, 24.8, 13.0, 5.2, and 2.6 Tg yr$^{-1}$ for SSP1-1.9 inventory (Gidden et al., 2019), respectively, and those are 23.7, 153.6, 28.5, 16.9, 10.5, 2.5, and 1.5 Tg yr$^{-1}$ for MEIC inventory (Zheng et al., 2018), respectively. The annual emissions of all pollutants over China in the SSP1-1.9 are higher than in the MEIC. However, compared with observations, the simulated seasonal mean MDA8 $O_3$ concentrations from the GEOS-Chem model with SSP1-1.9 inventory

have NMBs of 7.1-12.1% (Fig. 3), indicating that the model can capture fairly well the observations in China in 2015.

3. Line 360-361: The vertical profiles of present-day seasonal mean $O_3$ mass are shown in Figure S2, but not Figure 8.

**Response:**

Figure 8a shows the vertical profiles of present-day seasonal mean $O_3$ mass (Gg d$^{-1}$) for five processes (including net chemical production, PBL mixing, dry deposition, cloud convection, and horizontal and vertical advection transport), while Figure S2 shows the vertical profiles of present-day seasonal mean $O_3$ concentrations (ppbv). Therefore, the statement in text is correct.

4. The citation of figures and sections in texts should be carefully checked. For example, "Figure S3" should be "Figure S2" in Line 369; "Sect. 5.1" should be "Sect. 3.3.1".

**Response:**

Thanks for pointing this out. We carefully checked the citation of figures and sections in texts, and corrected the errors.

5. Figure S6: The tropospheric columns of $NO_2$ seem to be significantly overestimated by the models compared with the OMI satellite data. Reasonable attributions should be given and its impact on the FNR analysis should be discussed. Is it due to the uncertainties in the SSPs inventory? Besides, validations of simulated $NO_2$ near surfaces like Figure S1 using the surface measurements are suggested.

**Response:**

There are several possible reasons for the overestimation of tropospheric $NO_2$ columns in GCAP 2.0 simulation. Firstly, the uncertainties in the OMI products. Shah et al. (2020) reported that the GEOS-Chem tends to overestimate tropospheric $NO_2$ columns compared to the European Quality Assurance for Essential Climate Variables (QA4ECV) retrieval product (used in this work) (Boersma et al., 2018), owing to the strong sensitivity of the vertical distribution of $NO_2$ to that of aerosols and the misclassification of polluted scenes with high aerosol optical depth (and likely high $NO_2$) as clouds in QA4ECV retrieval. Secondly, SSP1-1.9 has higher $NO_x$ emissions in 2015 compared to MEIC. Finally, the inconsistencies in the sampling time. The GEOS-Chem model in this work only outputs the daily $NO_2$ values, while the overpass time of OMI satellite is about 13:45 local time. The observed tropospheric $NO_2$ column around 13:45 by Geostationary Environment Monitoring Spectrometer (GEMS) geostationary satellite was generally lower than the daily mean from the GEOS-Chem in Beijing for DJF 2021/22 and JJA 2022 (Yang et al., 2024).

To evaluate the impacts of this overestimation on the FNR analysis, we have examined the distributions of seasonal mean FNR over EC in 2015 from both the model and OMI observations (Figure R1 below). Compared to the model results, the observed VOC-limited regime shrinks toward the NCP and its surrounding areas in DJF, MAM, and SON, and in these seasons decreases in anthropogenic emissions increase MDA8 $O_3$ concentrations. As a result, the uncertainties in FNR do not affect our analysis of the effects of future emission changes (Figure 4).

Following the Reviewer's suggestion, we have added Figure S8 in the Supplementary Material to evaluate the model performance for surface $NO_2$. We have also added the following sentences in the S1 of Supplementary Material to describe the model performance: "We also compared the simulated surface $NO_2$ concentrations with observations from CNEMC in Fig. S8. The model generally captured the observed monthly variation in surface $NO_2$ concentrations in EC, NCP, and YRD, with R values of 0.44-0.70. The systematic low biases of surface $NO_2$ concentrations in the GEOS-Chem model (NMBs ranging from -51.7% to -19.2% in this work) were also reported in previous studies (Qu et al., 2020; Qu et al., 2022; Fang et al., 2024), because of the lack of representation of the spatial gradients in $NO_2$ observations within the coarse GEOS-Chem grid cells (Qu et al., 2022).".

[Figure]

Figure R1. Distributions of seasonal mean formaldehyde nitrogen ratio (FNR) over EC in 2015 from (a) the model and (b) the OMI observations.

[Figure]

Figure S8. (a)-(c) Monthly variations in simulated and observed surface $NO_2$ concentrations (ppbv) over (a) EC (with a total of 68 grids), (b) NCP (with a total of 6 grids), and (c) YRD (with a total of 4 grids) regions. Bars represent the range from first to third quartiles of all grid samples in this region. (d)-(f) The scatterplot of simulated versus observed monthly mean surface $NO_2$ concentrations for grids in EC, NCP, and YRD. The linear fit (black solid line and equation), correlation coefficient (R), and normalized mean biases (NMB) that calculated for grids in these three regions are also shown when all of the year 2015 data are considered.

**References:**

Boersma, K. F., Eskes, H. J., Richter, A., De Smedt, I., Lorente, A., Beirle, S., van Geffen, J. H. G. M., Zara, M., Peters, E., Van Roozendael, M., Wagner, T., Maasakkers, J. D., van der A, R. J., Nightingale, J., De Rudder, A., Irie, H., Pinardi, G., Lambert, J.-C., and Compernolle, S. C.: Improving algorithms and uncertainty estimates for satellite $NO_2$ retrievals: results from the quality assurance for the essential climate variables (QA4ECV) project, Atmos. Meas. Tech., 11, 6651-6678, https://doi.org/10.5194/amt-11-6651-2018, 2018.

Fang, L., Jin, J., Segers, A., Li, K., Xia, J., Han, W., Li, B., Lin, H. X., Zhu, L., Liu, S., and Liao, H.: Observational operator for fair model evaluation with ground $NO_2$ measurements, Geosci. Model Dev., 17, 8267-8282, https://doi.org/10.5194/gmd-17-8267-2024, 2024.

Gidden, M. J., Riahi, K., Smith, S. J., Fujimori, S., Luderer, G., Kriegler, E., van Vuuren, D. P., van den Berg, M., Feng, L., Klein, D., Calvin, K., Doelman, J. C., Frank, S., Fricko, O., Harmsen, M., Hasegawa, T., Havlik, P., Hilaire, J., Hoesly, R., Horing, J., Popp, A., Stehfest, E., and Takahashi, K.: Global emissions pathways under different socioeconomic scenarios for use in CMIP6: a dataset of harmonized emissions trajectories through the end of the century, Geosci. Model Dev., 12, 1443-1475, https://doi.org/10.5194/gmd-12-1443-2019, 2019.

Qu, Z., Henze, D. K., Cooper, O. R., and Neu, J. L.: Impacts of global $NO_x$ inversions on $NO_2$ and ozone simulations, Atmos. Chem. Phys., 20, 13109-13130, https://doi.org/10.5194/acp-20-13109-2020, 2020.

Qu, Z., Henze, D. K., Worden, H. M., Jiang, Z., Gaubert, B., Theys, N., and Wang, W.: Sector-Based Top-Down Estimates of $NO_x$, $SO_2$, and CO Emissions in East Asia, Geophys. Res. Lett., 49, https://doi.org/10.1029/2021gl096009, 2022.

Shah, V., Jacob, D. J., Li, K., Silvern, R. F., Zhai, S., Liu, M., Lin, J., and Zhang, Q.: Effect of changing $NO_x$ lifetime on the seasonality and long-term trends of satellite-observed tropospheric $NO_2$ columns over China, Atmos. Chem. Phys., 20, 1483-1495, https://doi.org/10.5194/acp-20-1483-2020, 2020.

Wang, Y., Liao, H., Chen, H., and Chen, L.: Future Projection of Mortality From Exposure to $PM_{2.5}$ and $O_3$ Under the Carbon Neutral Pathway: Roles of Changing Emissions and Population Aging, Geophys. Res. Lett., 50, https://doi.org/10.1029/2023gl104838, 2023.

Yang, L. H., Jacob, D. J., Dang, R., Oak, Y. J., Lin, H., Kim, J., Zhai, S., Colombi, N. K., Pendergrass, D. C., Beaudry, E., Shah, V., Feng, X., Yantosca, R. M., Chong, H., Park, J., Lee, H., Lee, W.-J., Kim, S., Kim, E., Travis, K. R., Crawford, J. H., and Liao, H.: Interpreting Geostationary Environment Monitoring Spectrometer (GEMS) geostationary satellite observations of the diurnal variation in nitrogen dioxide ($NO_2$) over East Asia, Atmos. Chem. Phys., 24, 7027-7039, https://doi.org/10.5194/acp-24-7027-2024, 2024.

Zheng, B., Tong, D., Li, M., Liu, F., Hong, C., Geng, G., Li, H., Li, X., Peng, L., Qi, J., Yan, L., Zhang, Y., Zhao, H., Zheng, Y., He, K., and Zhang, Q.: Trends in China's anthropogenic emissions since 2010 as the consequence of clean air actions, Atmos. Chem. Phys., 18, 14095-14111, https://doi.org/10.5194/acp-18-14095-2018, 2018.